# Synergy-of-Experts: Collaborate to Improve Adversarial Robustness

**Sen Cui[1*], Jingfeng Zhang[2*], Jian Liang[3], Bo Han[4], Masashi Sugiyama[2,5], Changshui Zhang[1]**

[1] Institute for Artificial Intelligence, Tsinghua University (THUAI),
Beijing National Research Center for Information Science and Technology (BNRist),
Department of Automation,Tsinghua University, Beijing, P.R.China
[2] RIKEN Center for Advanced Intelligence Project, Tokyo, Japan
[3] Alibaba Group, China
[4] Hong Kong Baptist University, Hong Kong SAR, China
[5] The University of Tokyo, Tokyo, Japan

`cuis19@mails.tsinghua.edu.cn`   `jingfeng.zhang@riken.jp`
`xuelang.lj@alibaba-inc.com`   `bhanml@comp.hkbu.edu.hk`
`sugi@k.u-tokyo.ac.jp`   `zcs@mail.tsinghua.edu.cn`

## Abstract

Learning adversarially robust models requires invariant predictions to a small neighborhood of its natural inputs, often encountering *insufficient model capacity*. There is research showing that learning multiple sub-models in an *ensemble* could mitigate this insufficiency, further improving the generalization and the robustness. However, the ensemble's voting-based strategy excludes the possibility that *the true predictions remain with the minority*. Therefore, this paper further improves the ensemble through a *collaboration* scheme—Synergy-of-Experts (SoE). Compared with the voting-based strategy, the SoE enables the possibility of correct predictions even if there exists a single correct sub-model. In SoE, every sub-model fits its specific vulnerability area and reserves the rest of the sub-models to fit other vulnerability areas, which effectively optimizes the utilization of the model capacity. Empirical experiments verify that SoE outperforms various ensemble methods against white-box and transfer-based adversarial attacks. The source codes are available at `https://github.com/cuis15/synergy-of-experts`.

## 1 Introduction

Deep models have been widely applied in various real-world applications including high-stakes scenarios (such as in healthcare, finance, and autonomous driving). An increasing concern is whether these models make *adversarially robust* decisions [1, 2]. Recently, there are research revealing that an adversarially robust method requires invariant predictions to a small neighborhood of its natural inputs, thus often encountering insufficient model capacity [3, 4]. This limits the further improvement of robustness and has the undesirable degradation of generalization [5].

Learning multiple sub-models in an ensemble [6, 7] can mitigate this insufficiency. Remarkably, there are research [8, 9, 10] proposing to minimize the vulnerability overlaps between each pair of sub-models and improving both robustness and generalization over a single model. However, the voting-based ensemble may waste the limited capacity of multiple models.

In the example of three sub-models (see Figure 1(b)), the adversarial input that lies in the black areas can fool the ensemble successfully, i.e., more than half of sub-models must correctly classify the adversarial input. Therefore, the ensemble's voting-based strategy excludes the possibility that *true*

---

*The first two authors have made equal contributions.

36th Conference on Neural Information Processing Systems (NeurIPS 2022).

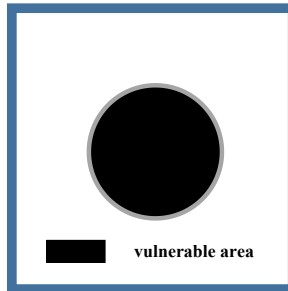
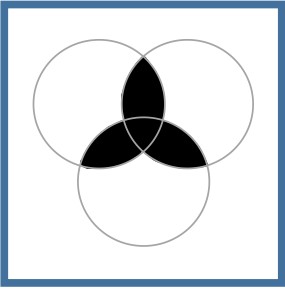
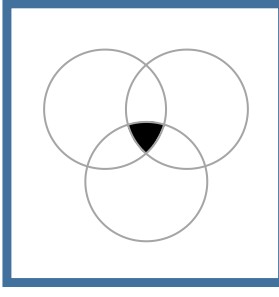

(a) Single model          (b) Ensemble          (c) Collaboration

Figure 1: Illustrations of the vulnerability area of (a) Single model (b) Ensemble, and (c) Collaboration. The black area represents the vulnerability area in which the model is undoubtedly fooled.

*predictions remain with the minority*. In other words, learning an ensemble requires more than half of the sub-models to fit the same vulnerability areas, which leaves the following question unanswered whether we could only leverage a single sub-model to fit a vulnerability area and reserve the rest of the sub-models to fit other vulnerability areas.

Inspired by mixture-of-experts (MoE) [11], we propose to *learn a collaboration* among multiple sub-models to optimally utilize the limited capacity. As shown in Figure 1(c), the adversarial input that lies in the vulnerability overlaps of all sub-models can undoubtedly fool the collaboration. Compared with the ensemble in Figure 1(b)), collaboration enables the possibility of correct predictions even if there exists a single correct sub-model. Besides, learning a collaboration could enable every sub-model to fit its vulnerability area, which could collectively fix broader vulnerability areas. Then, sub-models could collaboratively choose a trustworthy one to make the final predictions.

Classic MoE methods assume that the problem space is separable, and the separability is irrelevant to the learned model [11]. However, the non-i.i.d adversarial inputs [12] depend on the learned models and are hard to be classified by a learned gate in MoE. To tackle the above challenge, we propose Synergy-of-Experts (SoE), which explicitly builds the relationship between learned models and adversarial inputs. Specifically, each sub-model has dual heads: one outputs a vector of predicted probability $f_\theta(\cdot)$; another outputs a scalar that measures the confidence of the prediction. In the adversarial training phase, given an adversarial input $x$, each sub-model chooses an easy one(s) to feed itself. The other head is meanwhile updated by comparing the predicted probability on the true label—$f_\theta^y(\cdot)$ (a scalar). In the inference phase, given an input, SoE chooses a sub-model with the largest confidence as the representative to output the overall prediction.

We highlight our key contributions as follows.

- We provide a new perspective on learning multiple sub-models for defending against adversarial attacks. We show that the collaboration could make better decisions than the ensemble (Proposition 1), which implies collaboration may fix broader vulnerability areas.

- We propose a novel collaboration framework—SoE (see Section 3.2). In the training phase, SoE minimizes the vulnerability overlap of all sub-models; In the inference phase, SoE could effectively choose a representative sub-model to make correct predictions. We also provide a comprehensive analysis illustrating the rationale of SoE in Appendix.

- Empirical experiments corroborate the SoE outperforms various ensemble methods [10, 8, 9, 13] against white-box and transfer attacks.

## 2 Related Work

**Adversarial attack.** Adversarial attacks aim to craft the human-imperceptible adversarial input to fool the deep models. Adversarial attacks could be roughly divided into white-box attacks in which the adversary is fully aware of the model's structures [1, 14, 15, 16, 17, 18, 19, 20, 21, 22, 23, 24, 25, 26] and black-box attacks in which the deep models are treated as black boxes to the adversary [27, 28, 29, 30, 31, 32, 33, 34, 35, 36]. This paper focuses on building effective defense and select both white-box and black-box attack methods as our robustness evaluation metrics.

**Adversarial defense.** Defending adversarial attacks is a challenging task and researchers have proposed various solutions. *Certified defense* tries to learn provably robust deep models against norm-bounded (e.g., $\ell_2$ and $\ell_\infty$) perturbations [37, 38, 39, 40, 41, 42, 43, 44, 45, 46, 47, 48, 49]. *Empirical defense.* leverages adversarial data to build effective defense such as *adversary detection* [50, 51, 52, 53, 54, 55, 56, 57, 58, 59, 60, 61, 62, 63, 64, 65, 66, 67, 68] and *adversarial training* (AT), in which AT stands out as the most effective defense. Researchers have investigated various aspects of AT, such as improving AT's robustness or generalization [5, 69, 70, 71, 72, 73, 74, 75, 76, 77, 78, 79, 80, 81, 82, 83, 84, 85, 86, 87, 88, 89, 3], fixing AT's undesirable robust overfitting [90, 91, 92], improving AT's training efficiency [93, 94, 95, 96, 97, 98], understanding/interpreting AT's unique traits [99, 100, 101, 102, 103, 104, 105, 106, 107, 108, 49, 109, 110], applying AT into applications [111, 112], etc. Besides, researchers have also actively investigated robust-structured models [113, 114, 115, 116, 117, 118, 119]. Nevertheless, the above research thoroughly investigated a single model; this paper focuses on the collaboration among multiple models for adversarial defense.

**Ensemble methods for adversarial robustness.** The most relevant studies are the ensemble methods. Ensemble methods such as bagging [6] and boosting [7] have been investigated for significantly improving the model's generalization. Motivated by the benefits of ensemble methods in improving generalization, researchers introduced an ensemble to improve the model robustness [10, 9, 8, 120]. Tramèr *et al.* [120] proposed to reduce the adversarial transferability by training a single model with adversarial examples from multiple pretrained sub-models. Pang *et al.* [8] introduce a regularization method—ADP—to encourage high diversity in the non-maximal predictions of sub-models. Kariyappa *et al.* [9] improved the ensemble diversity by maximizing the introduced cosine distance between the gradients of sub-models with respect to the input. Yang *et al.* [10] proposed to distill non-robust features in the input and diversify the adversarial vulnerability. These methods reduced overlaps of vulnerability areas between sub-models [10]. Compared with voting strategy in ensemble, mixture-of-experts (MoE) assumes that the problem space can be divided into multiple sub-problems through a gate module [11, 121]. However, in adversarial training, the adversarial samples, which depend on the learned models, are not i.i.d. A vanilla MoE is hard to identify the best performing sub-models for each adversarial sample without the information about the learned models.

## 3 Collaboration to Defend Against Adversarial Attacks

### 3.1 Superiority of Collaboration

This section shows a *collaboration*, in theory, could make better decisions than an *ensemble*.

**Ensemble.** Suppose that there are $M$ learned sub-models $\{f_{\theta_1}, f_{\theta_2}, ..., f_{\theta_M}\}$, given an input $x$, $M$ sub-models make predictions $\{f_{\theta_1}(x), f_{\theta_2}(x), ..., f_{\theta_M(x)}\}$. The ensemble outputs a final prediction $\text{ensemble}(x, f_{\theta_1}, ..., f_{\theta_M})$ by the voting-based strategy:

$$\text{ensemble}(x, f_{\theta_1}, ..., f_{\theta_M}) = \arg\max_{y \in \{1,...,K\}} \left( \sum_{i=1}^{M} \mathbb{1}_{y = f_{\theta_i}(x)} \right), \tag{1}$$

where $\mathbb{1}$ is the indicator function and $K$ denotes the number of classes. Note that the ensemble outputs the predicted label $y$ that agrees with the majority predictions of the sub-models.

**Definition 1** (best-performing sub-model)**.** *Given an input $x$ and its label $y$, the best-performing sub-model achieves the lowest objective loss on the data $(x, y)$ among all $M$ sub-models:*

$$f_{\theta_{\text{best}}}(x) = \arg \min_{f_{\theta_i} \in \left\{ f_{\theta_1}, ..., f_{\theta_M} \right\}} \ell(f_{\theta_i}(x), y). \tag{2}$$

Note that the best-performing sub-model is w.r.t. the input data $(x, y)$, i.e., different input data correspond to different best-performing sub-models.

**Collaboration.** Suppose that there are $M$ learned sub-models $\{f_{\theta_1}, f_{\theta_2}, ..., f_{\theta_M}\}$. Given an input $x$, sub-models make predictions $\{f_{\theta_1}(x), f_{\theta_2}(x), ..., f_{\theta_M(x)}\}$. The collaboration tries to output a final prediction $\text{collaboration}(x, f_{\theta_1}, ..., f_{\theta_M})$ by the best-performing sub-model:

$$\text{collaboration}(x, f_{\theta_1}, ..., f_{\theta_M}) = f_{\theta_{\text{best}}}(x). \tag{3}$$

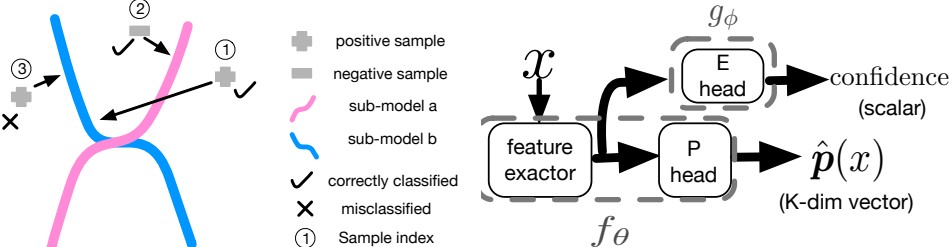

(a) Assign data to their best-performing sub-models     (b) Dual-head structured sub-models

Figure 2: (a) The blue and pink lines denote the decision boundaries of two sub-models. Each sub-model makes negative predictions $(-)$ on its left and makes positive predictions on its right $(+)$. The given data will be assigned to the sub-model that has the lowest objective loss. The arrows represent the data assignment. (b) Each sub-model has two heads—*P head* that outputs the predicted probability (vector) and *E head* approximates the predicted probability of the true label in the prediction (scalar).

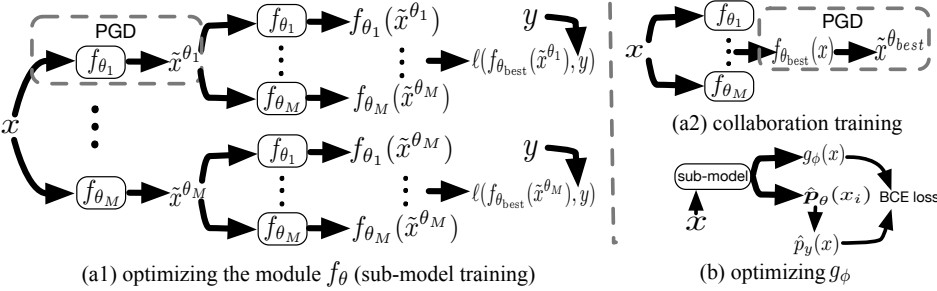

(a1) optimizing the module $f_\theta$ (sub-model training)     (b) optimizing $g_\phi$

Figure 3: Optimization process of the $M$ sub-models in a collaboration

**Proposition 1.** *Given $M$ learned sub-models, the predicted accuracy of the collaboration is upper-bounded that of the ensemble, i.e.,*

$$\mathbb{E}_{(x,y)\in D}\left[\mathbb{1}_{\text{collaboration}(x,f_{\theta_1},...,f_{\theta_M})=y}\right] \geq \mathbb{E}_{(x,y)\in D}\left[\mathbb{1}_{\text{ensemble}(x,f_{\theta_1},...,f_{\theta_M})=y}\right]. \qquad (4)$$

*Proof.* Given an $(x,y) \in D$, if the ensemble's prediction is correct, at least one sub-model makes correct prediction, i.e., $\mathbb{1}_{f_{\theta_{\text{best}}}(x)=y}$ holds; therefore, the collaboration' prediction is correct. If the collaboration's prediction is correct, there exists a case that the majority of sub-models make consistent but wrong predictions, while a single sub-model's prediction is correct; then, ensemble's prediction is wrong. Therefore, Proposition 1 holds. $\qquad\square$

From Proposition 1, a collaboration could achieve an equal or higher performance than an ensemble. Compared with the ensemble, the collaboration requires the identification of the best-performing sub-models using label information. Next, we will introduce a realization of our collaboration framework.

### 3.2 Realization of collaboration for defending against adversarial attacks

**Notation.** We firstly introduce the needed notations. Suppose $\mathcal{X}$ and $\mathcal{Y}$ denote input space and output space, where $\mathcal{Y} = \{1, ..., K\}$ for a $K$-class classification problem. There are $N$ samples in the dataset $D = \{(x,y)\}$, where $x \in \mathcal{X}$ and $y \in \mathcal{Y}$. Let $d_{\inf}(x,x') = \|x-x'\|_\infty$ denotes the infinity distance metric, and $\mathcal{B}_\epsilon[x] = \{x' \in \mathcal{X} \mid d_{\inf}(x,x') \leq \epsilon\}$ is the closed ball of of radius $\epsilon > 0$ centered at $x$. To search for adversarial data within norm ball $\mathcal{B}_\epsilon[x]$, [5] proposed a projected gradient descent (PGD) method that iteratively searches for adversarial data $\tilde{x}$ ($x$ refers to natural data). $f_\theta(x)$ outputs a K-dimensional predicted probability, i.e., $\hat{\boldsymbol{p}}(x) = [\hat{p}_1(x), ..., \hat{p}_K(x)]$.

**Goal of collaboration.** 1) ensure the correct prediction of the best-performing sub-model for a given input, and 2) select the best-performing sub-model among all sub-models to make predictions.

First, intuitively, every sub-model in a collaboration should maximize its expertise to fit its areas and leave the remaining areas fitted by others. As a result, the collaboration can minimize the vulnerability overlaps of all sub-models. Section 4 shows "minimizing the vulnerability overlap of all sub-models"

---

**Algorithm 1** training phase I: the sub-model training

---

**Input:** the sub-models with dual heads $\{f_{\theta_i}\}_{i=1}^{M}$ and $\{g_{\phi_i}\}_{i=1}^{M}$, where $f_{\theta_i}$ outputs the label prediction and $g_{\phi_i}$ outputs the approximated confidence, the training dataset $D$, and the hyperparameter $\sigma$

1: **for** each data $(x, y) \in D$ **do**
2:     **for** each sub-model $f_{\theta_i}, i = 1, 2, ..., M$ **do**
3:       Obtain the adversarial data $\tilde{x}^{\theta_i}$ of the sub-model $f_{\theta_i}$ using the PGD method;
4:       **for** each sub-model $f_{\theta_j}, j = 1, 2, ..., M$ **do**
5:         Calculate the approximated confidence, i.e., $g_{\phi_j}(\tilde{x}^{\theta_i})$;
6:         Minimize BCE loss $\ell_\phi = \mathrm{BCE}(g_{\phi_j}(\tilde{x}^{\theta_i}), \hat{p}_y(\tilde{x}^{\theta_i}))$ to update the module $g_{\phi_j}$;
7:         Collect sub-model $i$'s cross entropy (CE) loss on data $\tilde{x}^{\theta_i}$: $\ell\left(f_{\theta_j}\left(\tilde{x}^{\theta_i}\right), y\right)$;
8:       **end for**
9:       Calculate surrogate loss on data $\tilde{x}^{\theta_i}$: $\hat{\ell}_m = -\sigma \ln \sum_{j=1}^{M} \exp\left(\frac{-\ell\left(f_{\theta_j}\left(\tilde{x}^{\theta_i}\right), y\right)}{\sigma}\right)$;
10:       Update $\{f_{\theta_i}\}_{i=1}^{M}$ by minimizing $\hat{\ell}_m$. //choose the best-performing sub-model to fit $\tilde{x}^{\theta_i}$
11:     **end for**
12: **end for**
13: **Output:** the learned sub-models with dual heads $\{f_{\theta_i}\}_{i=1}^{M}$ and $\{g_{\phi_i}\}_{i=1}^{M}$.

---

is "minimizing the objective loss of the best-performing sub-models". Therefore, during the training phase, the given data should always be allocated to the sub-model that has the lowest objective loss. In other words, the sub-models always choose the easiest data to learn. In the example of Figure 2(a), $i$) Data③ is misclassified by both sub-models. The blue sub-model is near Data③ and has the lowest objective loss. We assign the blue sub-model to fit Data③. $ii$) Data② is correctly classified by the pink model but wrongly classified by the blue model; for ease of effort, we assign the pink model to fit Data②, because the collaboration can correctly be classified Data② by selecting the pink model as the representative. $iii$) Data① is correctly classified by both models. The blue model is far from Data① and takes the lowest effort on fitting it; therefore, we assign the blue model to fit Data①.

Second, to select the best-performing sub-model, we construct *dual-head structured sub-models*. As shown in Figure 2(b), our sub-model has dual heads: 1) *predictor* (P) head: $f_\theta$ predicts the label probability $f_\theta(x) = \hat{\boldsymbol{p}}(x) = [\hat{p}_1(x), ..., \hat{p}_K(x)]$ (a vector); 2) *evaluator* (E) head: $g_\phi$ approximates the confidence of the prediction $\hat{\boldsymbol{p}}(x)$.

Note that we use $g_\phi$ to approximate the true label probability $\hat{p}_y(x)$, which denotes the true confidence of a given prediction $\hat{\boldsymbol{p}}(x)$. Meanwhile, the largest confidence corresponds to the lowest objective loss, and vice versa (see theoretical proof in Proposition 2). Therefore, the best-performing sub-models could be identified using the approximated confidence ($g_\phi(x)$) by the E head.

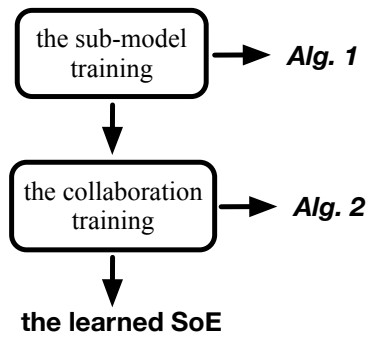

Figure 4: Training process of SoE.

To defend against adversarial attacks, the collaboration needs to learn from adversarial data. Algorithm 1,2 along with Figure 3 articulates how to learn such the collaboration. In particular, the training of our framework SoE shown in Figure 4 is as follows:

1. the sub-model training: the sub-models fit adversarial samples from their own or other sub-models;

2. the collaboration training: the sub-models fit adversarial samples from the collaboration.

**The sub-model training.** As shown in Figure 3(a1), given a natural training data $x$ and its label $y$, we use the PGD method to obtain $M$ adversarial variants $\{\tilde{x}^{\theta_i}\}_{i=1}^{M}$ of $M$ sub-models as existing baselines do. For each adversarial variant $\tilde{x}^{\theta_i}$, we assign the best-performing sub-model to learn and update its feature extractor and label head. This process corresponds to Lines 1–8 and Lines 9–10 in Algorithm 1. Note that in Line 9, we use a surrogate loss to approximate the sub-model assignment process (reasons see Eq. 5 and 6). As shown in Figure 3(b), given an adversarial variant $\tilde{x}^{\theta_i}$, we propose to use the binary-cross-entropy (BCE) loss between

---
**Algorithm 2** training phase II: the collaboration training
---
**Input:** sub-models with dual heads $\{f_{\theta_i}\}_{i=1}^{M}$ and $\{g_{\phi_i}\}_{i=1}^{M}$, where $f_{\theta_i}$ outputs the label prediction and $g_{\phi_i}$ outputs the approximated confidence, training dataset $D$, hyperparameter $\sigma$;

  1: **for** each data $(x, y) \in D$ **do**
  2:     **for** each sub-model $f_{\theta_i}, i = 1, 2, ..., M$ **do**
  3:         Calculate the approximated confidence, i.e., $g_{\phi_i}(x)$;
  4:         Calculate the prediction i.e., $f_{\theta_i}(x)$;
  5:     **end for**
  6:     Output the prediction $\hat{\boldsymbol{p}}'(x)$ with the highest confidence;
  7:     Obtain $\tilde{x}$ by perturbing $x$ to worsen the prediction; // generate the adversarial samples of the collaboration
  8:     Minimize BCE loss $\ell_\phi(\tilde{x})$ to update the module $g_\phi$ of all sub-models;
  9:     Update $\{f_{\theta_i}\}_{i=1}^{M}$ to fit $\tilde{x}$ by minimizing the surrogate loss $\hat{\ell}_m(\tilde{x})$;
10: **end for**
11: **Output:** the learned sub-models $\{f_{\theta_i}\}_{i=1}^{M}$ with $\{g_{\phi_i}\}_{i=1}^{M}$.
---

the predicted label probability on the true label (i.e., $\hat{p}_y(\tilde{x}^{\theta_i})$) and the approximated confidence (i.e., $g_\phi(\tilde{x}^{\theta_i})$) to update each sub-model's E head. This process corresponds to Lines 5–6 in Algorithm 1.

**The collaboration training.** During the sub-model training, we learn the most adversarial data from each sub-model using the sub-models performing best. The most adversarial samples cannot attack the collaboration successfully. However, there may exist harmful samples (which may not be the most adversarial for any sub-model) that are unexplored and can attack all sub-models. Therefore, the sub-model training may converge without a full exploration of the adversarial samples. (The experimental verification about this could be found in Appendix.) To defend these potential adversarial samples, we propose the collaboration training shown in Figure 3 (a2) and Algorithm 2. Firstly, we propose to generate the adversarial samples that can worsen the outputs of the collaboration. In particular, for each data sample $x$, we output the prediction $\hat{\boldsymbol{p}}(x)$ whose confidence $g_\phi(x)$ is the highest. We perturb $x$ to fool the prediction using PGD method. Then we minimize a surrogate loss to fit this adversarial data $\tilde{x}$. This process is detailly shown in Algorithm 2. We use Algorithm 1 and Algorithm 2 in sequence to train our collaboration. Algorithm 2 is proposed to explore the adversarial samples which could be not the most adversarial samples of any sub-model, but could fool all sub-models. In Algorithm 2, we attack the collaboration iteratively to obtain the adversarial samples. However, in each iteration we update the adversarial sample $\tilde{x}$, the best-performing sub-model could be different. For example, in the first iteration, given the input $x$, the best-performing sub-model is $f_{\theta_1}$. We obtain an adversarial sample $\tilde{x}'$ by attacking $f_{\theta_1}$. However, in the second iteration, given the input $\tilde{x}'$, the best-performing sub-model is another sub-model (e.g., $f_{\theta_2}$). We attack the best-performing sub-model $f_{\theta_2}$ to obtain the adversarial sample $\tilde{x}''$. Therefore, by attacking the collaboration following Algorithm 2, we could obtain the adversarial sample $\tilde{x}''$ which is not the most adversarial samples but could fool all sub-models and is unseen in Algorithm 1.

During the inference phase shown in Algorithm 3 in Appendix, once $M$ sub-models are properly learned, SoE chooses a representative sub-model whose confidence is the highest among all sub-models, and then outputs this sub-model's prediction.

### 3.3 Analyses of SoE

**Optimizing the best-performing sub-models.** We firstly show that minimizing the vulnerability overlap of all sub-models is equal to minimizing the objective loss of the best-performing sub-models. For ease of optimization of the best-performing sub-model, we provide a surrogate loss.

The vulnerability overlap of all sub-models refers to the set of adversarial data $(\tilde{x}, y)$ that are misclassified by all sub-models, i.e., all sub-models' objective loss is higher than a certain degree $\delta$: $\min_{\theta_i \in \{\theta_1, ..., \theta_M\}} \ell(f_{\theta_i}(\tilde{x}, y)) > \delta, \quad \tilde{x} \in \tilde{D}$, where $\tilde{D}$ denotes the vulnerability overlap of all sub-models.

To reduce the vulnerability overlap of all sub-models, we only need to reduce objective loss of a single model, which is equal to minimizing the loss of the best-performing sub-model, i.e.,

$$\min_{\{\theta_1, \theta_2, ..., \theta_M\}} \mathbb{E}_{(x,y) \in D} \left( \mathbb{E}_{i \in \{1,2,...,M\}} \ell_{best}(\tilde{x}^{\theta_i}, y) \right)$$

$$\text{where} \quad \ell_{best}(\tilde{x}^{\theta_i}, y) = \min_{j \in \{1,2,...,M\}} \ell \left( f_{\theta_j}(\tilde{x}^{\theta_i}), y \right)$$

(5)

where $\tilde{x}^{\theta_i}$ is the adversarial data generated by the sub-model $f_{\theta_i}$.

While directly performing the outer minimization in Eq.(5) may cause a trivial solution (e.g., there is only one optimized sub-model), for ease of the optimization of Eq.(5) (Corresponding to Lines 9–10 in Algorithm 1), we provide a surrogate objective as follows.

$$\min_{\{\theta_1, \theta_2, ..., \theta_M\}} \mathbb{E}_{(x,y) \in D} \left( \mathbb{E}_{i \in \{1,2,...,M\}} \hat{\ell}_m(\tilde{x}^{\theta_i}, y) \right),$$

(6)

where $\hat{\ell}_m(\tilde{x}^{\theta_i}, y) = -\sigma \ln \sum_{j=1}^{M} \exp \left( \frac{-\ell\left(f_{\theta_j}(\tilde{x}^{\theta_i}), y\right)}{\sigma} \right)$ and $\sigma > 0$ is a pre-defined hyper-parameter.

In Eq.(6), we approximate the objective $\min_{\theta_j \in \{\theta_1, \theta_2, ..., \theta_M\}} \ell \left( f_{\theta_j}(\tilde{x}^{\theta_i}), y \right)$ using a smooth surrogated maximum function due to

$$\min_{j \in \{1,2,...,M\}} \ell \left( f_{\theta_j}(\tilde{x}^{\theta_i}), y \right) - \delta \cdot \ln(M) \le \hat{\ell}_m(\tilde{x}^{\theta_i}), y) \le \min_{j \in \{1,2,...,M\}} \ell \left( f_{\theta_j}(\tilde{x}^{\theta_i}), y \right).$$

(7)

The proof of Eq.(7) is in Appendix.

**The best-performing sub-model has the highest confidence.** We show that a sub-model with the highest confidence achieves the minimum of the objective loss among all sub-models, i.e., the best-performing sub-model.

**Proposition 2.** *Given an input $x$, the sub-model that has the highest confidence corresponds to the best-performing sub-model, i.e.,*

$$\arg\max_{j \in \{1,2,...,M\}} \text{confidence}(f_{\theta_j}(x)) = \arg\min_{j \in \{1,2,...M\}} \ell(f_{\theta_j}(x), y).$$

(8)

where confidence corresponds to the probability of the true label in a given $\hat{p}(x)$. To learn the E head to approximate the confidence of $\hat{p}(x)$, we could simply compare its output with the predicted probability on the true label (i.e.,$\hat{p}_y(x)$), then update the E head by gradient descent (corresponds to Lines 6–7 in Algorithm 1). Given an input to the collaboration, our dual-head structured sub-models can collaboratively decide the best-performing sub-model by comparing the values of the E head (corresponds to Algorithm 3 in Appendix).

Note that the E head may be susceptible to adversarial attacks in the white-box setting. In our implementation, we use a simple linear structure to regress the confidence. Experimental results on adaptive attacks demonstrate the reliability of our framework.

# 4 Experiments

In this section, we conduct experiments on a benchmark dataset to verify the effectiveness of our method in defending against white-box and transfer attacks. Then we provide ablation studies to demonstrate the significance of the collaboration training described in Algorithm 2. The experimental results about black-box attacks and more discussions could be found in Appendix.

## 4.1 Experimental Setup

Following the work in [10], we compare our method with various related methods, including ADP [8], GAL [9], DVERGE [10] and MoRE [13]. We use ResNet-20 [122] as sub-models in all methods for fair comparisons, and we use CIFAR10 as the data set, a classical image dataset [123] that has 50,000 training images and 10,000 test images.

## 4.2 Performance on White-box attack

As there are mainly two threat modes in the adversarial attack setting: white-box attack and black-box attack. White-box attack refers to that attackers know all the information about the models, including training data, model architectures, and parameters, while black-box attackers have no access to the information about the model's structures and parameters and rely on surrogate models to generate transferable adversarial examples.

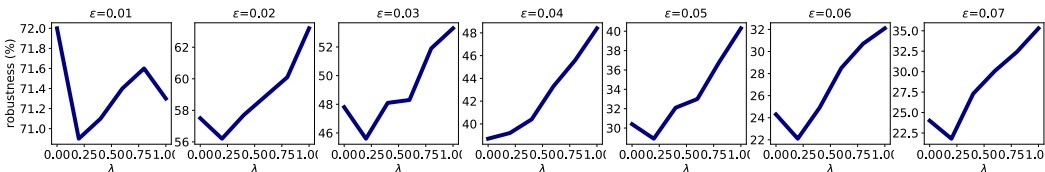

Figure 5: SoE robustness with varying $\lambda$ under adaptive attacks with $\epsilon \in \{0.01, 0.02, ..., 0.07\}$.

We compare our method with 4 baselines on defending against white-box attacks using a subset of **CIFAR10**. We use 50-step PGD with five random starts and the step size of $\epsilon/5$ to attack all methods as in [10]. We learn the models in adversarial training and evaluate the robustness under various attacks with the same $\epsilon$. For example, to evaluate the robustness under white-box attack with $\epsilon = 0.01$, we first learn the models in adversarial training with $\epsilon = 0.01$. We evaluate all methods following the setting in [10]. In particular, we randomly select 1000 samples under different $\epsilon$. For the PGD attack, we select the cross-entropy loss to update the perturbations to search for adversarial samples. In addition to the robustness, we also report the performance of all methods on clean data with the adversarial training under different $\epsilon$.

Table 1: Robustness and clean data accuracy (%) under white-box attack.

| $\epsilon$(robust/clean) | 0.01 | 0.02 | 0.03 | 0.04 | 0.05 | 0.06 | 0.07 |
|---|---|---|---|---|---|---|---|
| GAL | 49.5/87.8 | 31.4/85.4 | 25.4/81.2 | 22.7/78.7 | 18.4/77.3 | 13.4/76.2 | 9.0/76.0 |
| DVERGE | 67.3/85.4 | 52.3/83.0 | 41.1/79.7 | 29.9/77.6 | 22.5/76.7 | 14.2/75.8 | 10.0/75.3 |
| MoRE | 67.9/88.0 | 49.9/85.3 | 37.8/82.0 | 31.3/79.5 | 24.0/78.2 | 15.6/77.1 | 12.3/77.8 |
| ADP | 67.7/89.0 | 52.9/86.8 | 40.8/85.4 | 30.8/83.3 | 25.8/76.0 | 23.4/66.4 | 20.3/63.0 |
| SoE | **72.0**/88.8 | **57.5**/85.6 | **47.8**/80.2 | **38.7**/80.0 | **30.4**/79.1 | **24.3**/76.7 | **24.0**/74.1 |
| SoE (adaptive) | 70.9/88.8 | 56.2/85.6 | 45.6/80.2 | 38.7/80.0 | 28.9/79.1 | 22.1/76.7 | 21.8/74.1 |

From Table 1, SoE achieves a better robustness performance under white-box attack. The results verify that collaboration significantly improves the utilization of the limited model capacity. Therefore, SoE can fit more adversarial data and have a relatively smaller vulnerable area.

**Performance on Adaptive Attacks.** We follow the suggestions in [124] and conduct two different adaptive attacks to fool the dual heads simultaneously. For the first adaptive attack, we attack the E head to minimize the confidence of the best-performing sub-model. In particular, we maximize $l_1 = \mathrm{BCE}(g_{\phi_j}(x), 1)$, where $j = \arg\max_{i \in [M]} g_{\phi_i}(x)$ means the $j$-th sub-model is identified as the best-performing sub-model. For the second adaptive attack, we try to achieve a mismatch between the correct predictions and the highest confidence. Specifically, we maximize $l_2 = -\log\left[f_{\theta_j}(x)_y * g_{\phi_j}(x) + 1 - g_{\phi_j}(x)\right]$, where $j = \arg\max_{i \in [M]} -\log\left[f_{\theta_i}(x)_y * g_{\phi_i}(x) + 1 - g_{\phi_i}(x)\right]$. We conduct experiments by maximizing the weighted loss $\ell_1^{adp} = \ell(f_\theta(x), y) + \lambda \cdot l_1$ and $\ell_2^{adp} = \ell(f_\theta(x), y) + \lambda \cdot l_2$ with varying $\lambda$. The robustness with respect to $\lambda$ on the stronger adaptive attack (i.e., the attack achieves a higher success rate) is shown in Figure 5. Though both adaptive attacks could attack the predictor and the evaluator

Table 2: Transfer attack with 3 adversarial variants (%).

| $\epsilon$ methods | 0.01 | 0.02 | 0.03 | 0.04 | 0.05 | 0.06 | 0.07 |
|---|---|---|---|---|---|---|---|
| GAL | $64.2_{\pm 4.2}$ | $48.7_{\pm 2.7}$ | $50.2_{\pm 3.5}$ | $49.9_{\pm 3.2}$ | $52.3_{\pm 4.5}$ | $48.7_{\pm 3.2}$ | $42.2_{\pm 4.1}$ |
| ADP | $\mathbf{85.6}_{\pm .2}$ | $82.9_{\pm .2}$ | $78.3_{\pm .3}$ | $73.2_{\pm .1}$ | $69.6_{\pm .2}$ | $60.4_{\pm .2}$ | $57.4_{\pm .1}$ |
| MoRE | $84.8_{\pm .3}$ | $82.1_{\pm .1}$ | $78.4_{\pm .2}$ | $74.3_{\pm .1}$ | $73.2_{\pm .1}$ | $70.3_{\pm .2}$ | $69.1_{\pm .3}$ |
| DVERGE | $83.4_{\pm .3}$ | $80.1_{\pm .2}$ | $77.3_{\pm .1}$ | $72.4_{\pm .1}$ | $71.9_{\pm .2}$ | $68.8_{\pm .3}$ | $66.2_{\pm .2}$ |
| SoE | $85.2_{\pm .1}$ | $\mathbf{83.4}_{\pm .1}$ | $\mathbf{78.8}_{\pm .1}$ | $\mathbf{76.6}_{\pm .2}$ | $\mathbf{74.6}_{\pm .1}$ | $\mathbf{72.3}_{\pm .2}$ | $\mathbf{70.2}_{\pm .2}$ |

simultaneously, the evaluator in our method is robust to the adversarial samples because of its sample structure. As seen in Figure 5 and Table 1, our method is slightly degraded by adaptive attacks and still outperforms baseline models.

### 4.3 Visualization of the our Collaboration Scheme

To intuitively understand the collaboration mechanism of our proposed SoE, we show the decision boundaries of the ensemble and the collaboration in Figure 6. In particular, we learn the ensemble and the collaboration with 3 ResNet-20 sub-models on CIFAR10 dataset with adversarial training.

Table 3: Transfer attack with 30 adversarial variants (%).

| $\epsilon$ methods | 0.01 | 0.02 | 0.03 | 0.04 | 0.05 | 0.06 | 0.07 |
|---|---|---|---|---|---|---|---|
| GAL | $57.8_{\pm 3.2}$ | $64.1_{\pm 3.7}$ | $46.3_{\pm 2.8}$ | $56.0_{\pm 3.0}$ | $43.9_{\pm 3.1}$ | $44.5_{\pm 2.9}$ | $41.4_{\pm 3.2}$ |
| ADP | $\mathbf{84.2}_{\pm .2}$ | $80.2_{\pm .2}$ | $73.7_{\pm .1}$ | $69.4_{\pm .2}$ | $65.2_{\pm .2}$ | $56.7_{\pm .2}$ | $54.4_{\pm .2}$ |
| MoRE | $83.7_{\pm .3}$ | $79.6_{\pm .2}$ | $74.4_{\pm .2}$ | $70.2_{\pm .3}$ | $67.6_{\pm .1}$ | $63.8_{\pm .3}$ | $59.3_{\pm .2}$ |
| DVERGE | $81.5_{\pm .3}$ | $78.1_{\pm .3}$ | $73.5_{\pm .3}$ | $68.4_{\pm .1}$ | $67.2_{\pm .2}$ | $63.8_{\pm .1}$ | $57.1_{\pm .2}$ |
| SoE | $83.1_{\pm .3}$ | $\mathbf{80.4}_{\pm .2}$ | $\mathbf{75.1}_{\pm .3}$ | $\mathbf{70.8}_{\pm .2}$ | $\mathbf{69.0}_{\pm .2}$ | $\mathbf{64.0}_{\pm .3}$ | $\mathbf{61.1}_{\pm .2}$ |

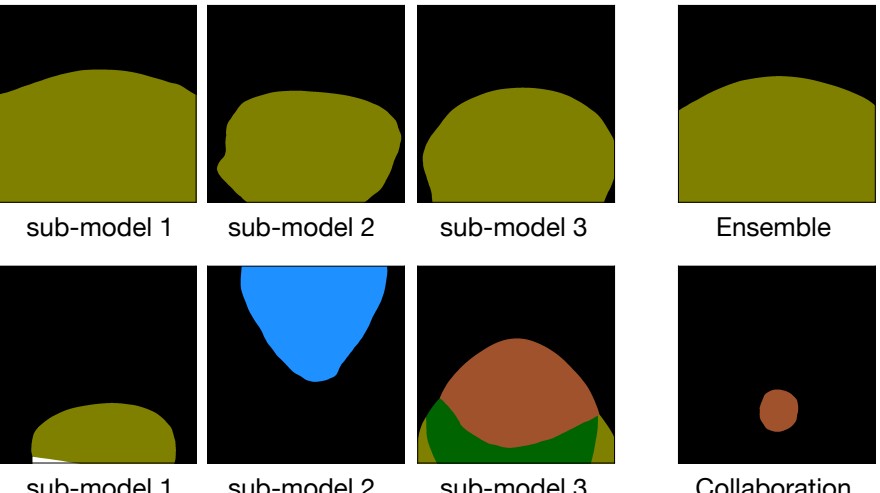

Figure 6: The visualization of the decision boundaries of the ensemble and the collaboration. The same color areas denotes the same predicted label. The black regions means that the models predict correctly while other color regions means that the models are fooled and predict incorrectly. The vertical axis is along the adversarial direction and the horizontal axis is along a random Rademacher vector.

The same color areas in Figure 6 denotes the same predicted label. The black regions means that the models predict correctly while other color regions means that the models are fooled and predict incorrectly. The ensemble defending against attacks requires more than half of the sub-models fit the same vulnerability areas. From the top of Figure 6, all sub-models fit similar vulnerability areas and there is a boarder vulnerability overlap between sub-models. Our proposed collaboration proposes to minimize the overlap of all sub-models. From the bottom of Figure 6, different sub-models defend against different vulnerability areas collaboratively and our collaboration achieves a smaller vulnerability areas.

## 4.4 Performance on Transfer Attack

Due to the transferability of adversarial examples, transfer adversaries can craft adversarial examples based on surrogate models and perform an attack on the target model. In our experiments, we follow the transfer attack setting in [10] and select 1000 test samples randomly. We use hold-out baseline ensembles with three ResNet-20 sub-models as the surrogate models to generate adversarial samples. In particular, we use three attack methodologies: PGD with momentum [35], SGM [29] which adds weight to the gradient through the skip connections of the model, and M-FGSM [36] which randomly augments the input images in each step. For each sample, three adversarial variants are using the three attack methods. Only when the model can classify all kinds of adversarial variants can the model successfully defend against adversarial attacks. We show the results of all methods in Table 2. Furthermore, we also use a more challenging setting following the work in [10]. We use hold-out baseline models with 3, 5, and 8 ResNet-20 sub-models as the surrogate models. Meanwhile, we generate adversarial samples with cross-entropy loss and CW loss [15]. For each sample, we generate 30 adversarial variants, and only if the model classifies all the 30 variants can the model defend the transfer attack successfully. The results are shown in Table 3.

In our experiments, GAL is hard to learn stably in adversarial training. In Table 2, when $0.01 \leq \epsilon$, SoE achieves a better performance compared with baselines. With the increase of $\epsilon$, the volume

of $\epsilon$-ball increases exponentially. The performances of all methods get worse significantly because of insufficient model capacity. Similar results can also be found in Table 3. Since SoE addresses more adversarial data using a collaboration mechanism, it achieves a relatively better robustness performance as $\epsilon$ increases.

### 4.5 The Robustness of SoE under Different Number of Sub-models

To explore the robustness of the collaboration under different $\epsilon$ with different number of sub-models, we conduct experiments under transfer attacks with different number of sub-models ($1 \leq N \leq 5$). The detailed information could be found in Section 4.3 in Appendix. In summary, we have the following findings. For different $\epsilon$, more sub-models could achieve a more significant robustness improvement with the increase of $\epsilon$. For different number of sub-models, more sub-models are more likely to achieve a higher robustness, but the margin gain decreases with more sub-models.

## 5 Conclusion

In this paper, we study an essential question in the field of adversarial attacks that when we should collaborate. ($i$) If a single model can handle everything, there is no need for multiple models. ($ii$) If a single model can only handle a part of the whole, collaboration among multiple models makes sense. Adversarial defense is a typical task that falls into the circumstance ($ii$) because a single model hardly fits adversarial data. We provided a collaboration framework—SoE—as the defense strategy over ensemble methods, and empirical experiments indeed verified the efficacy of SoE. Future work includes applying our collaboration framework to other areas such as kernel methods, fairness, and federated model, etc.

## Acknowledgments

We would like to thank the anonymous reviewers of NeurIPS 2022 for their constructive comments. Sen Cui and Changshui Zhang would like to acknowledge the funding by the Natural Science Fundation of China(NSFC. No. 62176132 ). Bo Han was supported by the RGC ECS No. 22200720, NSFC YSF No. 62006202, Guangdong Basic and Applied Basic Research Foundation No. 2022A1515011652, and RIKEN Collaborative Research Fund. Jingfeng Zhang was supported by JST ACT-X Grant Number JPMJAX21AF and JSPS KAKENHI Grant Number 22K17955, Japan. Masashi Sugiyama was supported by JST AIP Acceleration Research Grant Number JPMJCR20U3 and the Institute for AI and Beyond, UTokyo, Japan.

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
