# Appendix of Synergy-of-experts

## 1 Theoretical Proofs

The proof of Eq.(7) is as follows.

*Proof.* Since we have

$$\max_{j} \exp\left(\frac{-\ell\left(f_{\theta_j}(\tilde{x}^{\theta_i}), y\right)}{\sigma}\right) \leq \sum_{j=1}^{M} \exp\left(\frac{-\ell\left(f_{\theta_j}(\tilde{x}^{\theta_i}), y\right)}{\sigma}\right) \leq M \cdot \max_{j} \exp\left(\frac{-\ell\left(f_{\theta_j}(\tilde{x}^{\theta_i}), y\right)}{\sigma}\right),$$ 
$$(1)$$

considering that $-\delta \ln(x)$ monotonically decreases w.r.t $x$, we have

$$-\delta \ln\left(M \cdot \max_{j} \exp\left(\frac{-\ell\left(f_{\theta_j}(\tilde{x}^{\theta_i}), y\right)}{\sigma}\right)\right) \leq \hat{\ell}_m(f_{\theta_j}(\tilde{x}^{\theta_i}, y)) \leq -\delta \ln \max_{j} \exp\left(\frac{-\ell\left(f_{\theta_j}(\tilde{x}^{\theta_i}), y\right)}{\sigma}\right),$$
$$(2)$$

where $-\delta \ln\left(M \cdot \max_{j} \exp\left(\frac{-\ell\left(f_{\theta_j}(\tilde{x}^{\theta_i}), y\right)}{\sigma}\right)\right) = \min_{\theta_j \in \{\theta_1, \theta_2, ..., \theta_M\}} \ell\left(f_{\theta_j}(\tilde{x}^{\theta_i}), y\right) - \delta \cdot \ln(M)$

and $-\delta \ln \exp\left(\max_{j} \frac{-\ell\left(f_{\theta_j}(\tilde{x}^{\theta_i}), y\right)}{\sigma}\right) = \min_{\theta_j \in \{\theta_1, \theta_2, ..., \theta_M\}} \ell\left(f_{\theta_j}(\tilde{x}^{\theta_i}), y\right)$, so Eq.(7) holds. $\square$

The proof of Proposition 2 is as follows.

*Proof.* The objective loss depends on the label probability (e.g., cross-entropy loss $\ell = -\ln p_y(x)$), and the loss monotonically decreases with respect to $p_y(x)$, so the best-performing sub-model has the highest label probability. Combined the fact that $\mathrm{confidence}(\boldsymbol{p}(x))$ monotonically increases w.r.t. $p_y(x)$, the best-performing sub-model corresponds to the highest confidence; therefore, Eq.(8) holds. $\square$

## 2 Algorithms

The whole pipeline of our method during inference is summarized in Algorithm 1. For an input natural (or adversarial) sample, each sub-model outputs the prediction using P head and the confidence using E head. Then the collaboration predicts the label as the prediction with the highest confidence.

## 3 More Discussions about SoE

### 3.1 the Behavior of SoE on Synthetic Data

To demonstrate the behavior of collaboration, we apply our collaboration framework on the well-known **XOR** problem as an example.

In **XOR** problem, there is a binary training set $D = \{x_i, y_i\}_{i=1}^{n}$, where $y_i \in \{\pm 1\}$. For the samples with the label $y_i = 1$, the input feature $x$ is independently sampled from the two Gaussian distributions $x \in \mathcal{N}(\boldsymbol{\mu} = [1, 1], \boldsymbol{\sigma} = 0.1 \cdot \boldsymbol{I}_2)$ and $x \in \mathcal{N}(\boldsymbol{\mu} = [-1, -1], \boldsymbol{\sigma} = 0.1 \cdot \boldsymbol{I}_2)$, where $\mathcal{N}(\boldsymbol{\mu} = [1, 1], \boldsymbol{\sigma} = 0.1 \cdot \boldsymbol{I}_2)$ denotes 2-d Gaussian distribution with the mean vector $\boldsymbol{\mu} = [1, 1]$ and

**Algorithm 1** SoE (inference phase)

---

**Input:** the learned sub-models $\{f_{\theta_i}\}_{i=1}^M$ with $\{g_{\phi_i}\}_{i=1}^M$, test input $x$.

1: **for** all sub-models $f_{\theta_i}, i = 1, ..., M$ (in parallel) **do**
2:     make label predictions $\hat{\boldsymbol{p}}(x) = f_{\theta_i}(x)$ and output approximated confidence $g_{\phi_i}(x)$;
3: **end for**
4: **Return:** prediction $\hat{\boldsymbol{p}}(x)$ whose confidence $g_\phi(x)$ is the highest among M sub-models.

---

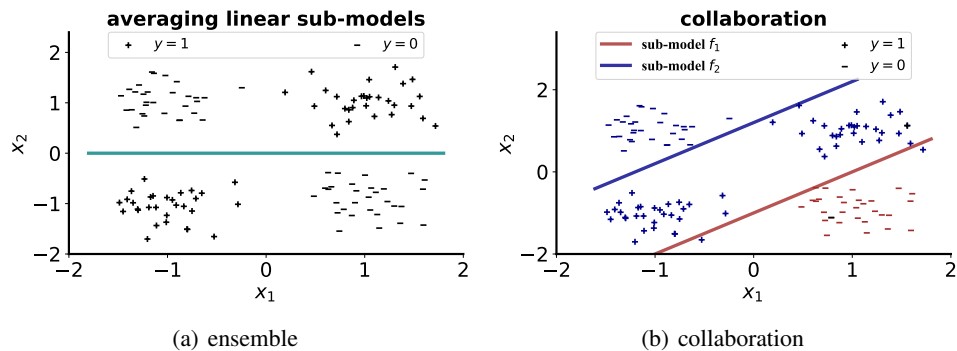

(a) ensemble               (b) collaboration

Figure 1: Illustration of the behavior of the ensemble and the collaboration.

covariance matrix $\boldsymbol{\sigma} = 0.1 \cdot \boldsymbol{I}_2$. For the samples with the label $y_i = -1$, we sample the input feature independently $x_i \in \mathcal{N}(\boldsymbol{\mu} = [-1, 1], \boldsymbol{\sigma} = 0.1 \cdot \boldsymbol{I}_2)$ and $x_i \in \mathcal{N}(\boldsymbol{\mu} = [1, -1], \boldsymbol{\sigma} = 0.1 \cdot \boldsymbol{I}_2)$.

Suppose there are two linear sub-models $f_1(x) = \boldsymbol{a}_1 \cdot x + b_1$ and $f_2(x) = \boldsymbol{a}_2 \cdot x + b_2$. Ensemble methods output a prediction $\hat{\boldsymbol{p}}(x)$ by a voting-based strategy e.g., averaging the predictions as the output, i.e., $\hat{\boldsymbol{p}}(x) = \frac{1}{M} \cdot \sum_{i=1}^M f_{\theta_i}(x)$.

Collaboration aims to 1). minimize the mean square error (MSE) of the best-performing sub-models; 2). optimize the module $g_1$ and $g_2$ to measure the quality of the multiple predictions during inference.

From Figure 1(a), learning multiple linear sub-models and averaging the predictions (ensemble) is still a linear model, so it cannot tackle **XOR** problem. Collaboration can address **XOR** which classifies each sample by learning to specify the sub-tasks to different sub-models. From Figure 1(b), the samples are assigned to two sub-models, in which the blue samples are assigned to $f_1$ and the red samples are assigned to $f_2$. Finally, it output the prediction by identifying the best-performing sub-models.

## 3.2 Optimization Efficiency

The training cost is a notable issue. We compare the training cost of all methods from the two aspects; 1). parameters and GFLOPs: all methods have the same model architecture (ResNet20), so all methods have a similar number of parameters and GFLOps. Compared with baselines, our method has an additional head (E head), which is a one-layer MLP with 128 parameters and has a negligible computation cost; 2). training manner; all methods except DVERGE achieve adversarial training by generating adversarial samples using PGD attack. The time consumption of all methods using the device Geforce 2080Ti (100 epochs) is in Table 1.

Table 1: time consumption of all methods (100 epoches).

| methods | GAL | DEVRGE | ADP | MoRE | SoE |
|---|---|---|---|---|---|
| time | 6 h 36min | 11 h 40 min | 6 h 35 min | 7 h 01min | 7 h 23 min |

DVERGE distills non-robust features by computing transferable adversarial samples, which have an O($N^2$) time complexity in which N is the number of sub-models, so it has a relatively large time consumption. Our method outperforms baselines by training an additional E head and it could cause an additional small-time consumption as shown in the above table.

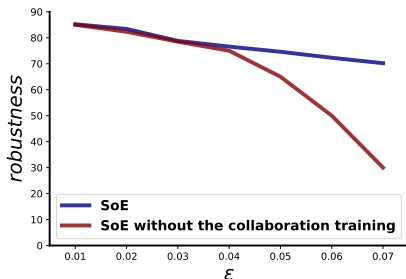
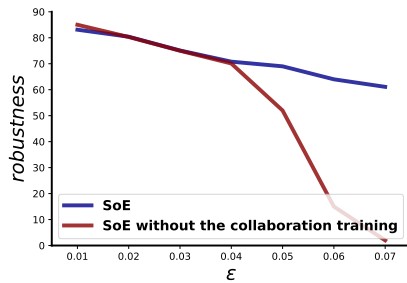

(a) attacks with 3 adversarial variants        (b) attacks with 30 adversarial variants

Figure 2: Robustness (%) on transfer attacks of two methods: 1) SoE; 2) SoE without the collaboration training.

### 3.3 a Single Big Model or a *Collaboration*

**advantages of the SoE**    Compared with SoE, a single big model may be difficult to fit the adversarial data. For a big single model with a deeper structure, it may face gradient vanish or gradient explosion during optimization. Without a well-designed optimization method, it could be more vulnerable to an imperceptible perturbation compared to a simple model. The SoE alleviates this problem by learning multiple relatively small models.

**disadvantages of the SoE**    A single big model may be a direct solution for addressing insufficient capacity in adversarial training. Compared with learning a single big model, SoE needs to design a complicated mechanism to improve the utilization of the model capacity.

### 3.4 SoE in Adversarial Training Can Avoid a Trivial Optimization

As our proposed collaboration mechanism enhances the performance of the best-performing sub-model. One may wonder whether it obtains a trivial case, e.g., only one sub-model is properly trained. In fact, our SoE does not bring such a trivial case. Fitting the adversarial data consumes a tremendous model capacity so the model usually cannot fit all adversarial samples. For a learned sub-model, there still exist adversarial samples that cannot handle. From our proposed collaboration mechanism, these adversarial samples generated from a learned sub-model are more likely to be assigned to other sub-models which perform better. Therefore, in our SoE, all sub-models will be trained.

Table 2: the accuracy (%) on the clean data

| SoE | sub-model A | sub-model B | sub-model A |
|-----|-------------|-------------|-------------|
| 85.6 | 83.9 | 84.2 | 83.9 |

To experimentally verify this claim, we present the accuracy on clean data of all three sub-models with adversarial training ($\epsilon = 0.02$) in the Table 2. From the Table 2, all sub-models have a similar performance on the clean data.

## 4 More Experimental Results and Analysis

### 4.1 Ablation Studies

To make full use of the limited modal capacity of an ensemble, SoE proposes the sub-model training (shown in Algorithm 1), which makes the adversarial samples from sub-models be learned by the best-performing ones. The sub-model training enables the most adversarial attacks of sub-models could be successfully defended. However, learning from the most adversarial samples of all sub-models could converge without a full exploration of the vulnerable area. Therefore, we propose the collaboration training (shown in Algorithm 2), which fits the adversarial samples unexplored before.

To verify the significance of the collaboration training, we conduct ablation studies under two transfer attacks. In particular, we train two kinds of models to defend against the attacks: 1). SoE ; 2). SoE without the collaboration training. The two transfer attacks are generated as stated in Section 4.4, 1). attacks with 3 adversarial variants; 2). attacks with 30 adversarial variants.

We show the robustness of two methods in Figure 2. From Figure 2(a) and 2(b), when $0.01 \leq \epsilon \leq 0.04$, SoE without the collaboration training achieves a similar robustness compared with SoE. It means that the sub-model owns relatively sufficient model capacity and the adversarial samples could be largely explored by the sub-model training. However, when $\epsilon \geq 0.04$, the robustness of SoE without the collaboration training degrades significantly. In this case, the need for model capacity increases dramatically. The most adversarial samples identified by the sub-model training cannot cover all cases, and our proposed the collaboration training mitigates it effectively.

## 4.2 Performance on Black-box Attack

In addition to transfer attack, query attack can also craft adversarial samples based on the predicted scores of the model. To evaluate the robustness of SoE under query attack, we use Square Attack method [1] to attack all methods. Square Attack selects localized square-shaped updates at random positions in each step [1]. In particular, we set $0.01 \leq \epsilon \leq 0.07$ and learn the models for each method with adversarial training. Then we randomly select 1000 samples from the test set and evaluate the robustness of each method under Square Attack with 5000 iterations. We repeated the experiments under 5 different random seeds and the results are presented in Table 3.

Table 3: Robustness results under Square Attack (%).

| methods \ $\epsilon$ | 0.01 | 0.02 | 0.03 | 0.04 | 0.05 | 0.06 | 0.07 |
|---|---|---|---|---|---|---|---|
| GAL | $47.0_{\pm 2.3}$ | $43.8_{\pm 1.8}$ | $26.4_{\pm 3.2}$ | $27.7_{\pm 2.1}$ | $18.5_{\pm 1.5}$ | $13.1_{\pm 1.2}$ | $8.40_{\pm 2.4}$ |
| ADP | $72.5_{\pm 1.0}$ | $60.3_{\pm 1.1}$ | $47.2_{\pm 1.3}$ | $37.9_{\pm 1.4}$ | $28.0_{\pm 1.3}$ | $25.5_{\pm 1.0}$ | $\mathbf{21.3}_{\pm 1.2}$ |
| MoRE | $72.8_{\pm .8}$ | $59.6_{\pm .8}$ | $46.4_{\pm 1.1}$ | $37.8_{\pm 1.2}$ | $30.1_{\pm 1.3}$ | $22.0_{\pm 1.7}$ | $16.6_{\pm 1.9}$ |
| DVERGE | $72.0_{\pm 1.2}$ | $58.8_{\pm 1.1}$ | $47.8_{\pm 1.0}$ | $37.9_{\pm 1.1}$ | $28.4_{\pm 1.0}$ | $21.0_{\pm 1.2}$ | $15.0_{\pm 1.1}$ |
| SoE | $\mathbf{77.7}_{\pm .5}$ | $\mathbf{65.1}_{\pm 1.0}$ | $\mathbf{54.7}_{\pm 1.0}$ | $\mathbf{45.3}_{\pm 1.0}$ | $\mathbf{36.3}_{\pm 1.3}$ | $\mathbf{29.2}_{\pm 1.3}$ | $18.0_{\pm 1.5}$ |

From Table 3, SoE outperforms the baselines under query attack. SoE optimizes the utilization of the model capacity by specifying each sub-model to handle the "specific" adversarial attacks, which defends against query attacks more efficiently. In addition, SoE has a different prediction mechanism which facilitates its robustness when defending against square attack. Baselines output the predicted labels by averaging the predictions. SoE outputs the labels by selecting the optimal representative to output the overall prediction. When black-box queries generate different adversarial noise for $x$, SoE could select different optimal sub-model which avoids stronger attacks, so SoE is harder to be fooled by Square attack which crafts attacks according to the predicted scores.

## 4.3 Analysis about the Number of Sub-models

To explore the effect of the number of sub-models, we conduct experiments under transfer attacks with different number of sub-models ($1 \leq N \leq 5$). The results are shown in the following Figure 3 and Figure 4.

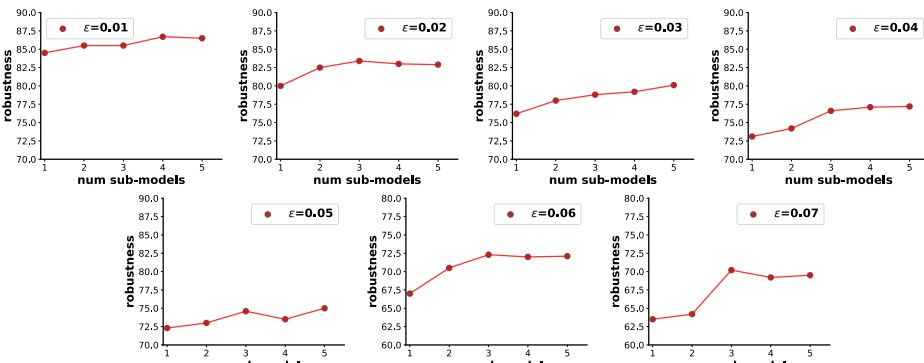

Figure 3: Robustness (%) using different number of sub-models under transfer attacks with 3 adversarial variants

From Figure 3, as $\epsilon = 0.01$, multiple sub-models achieve a similar adversarial robustness compared with a single model. In this case, a single has enough model capacity and the collaboration is not

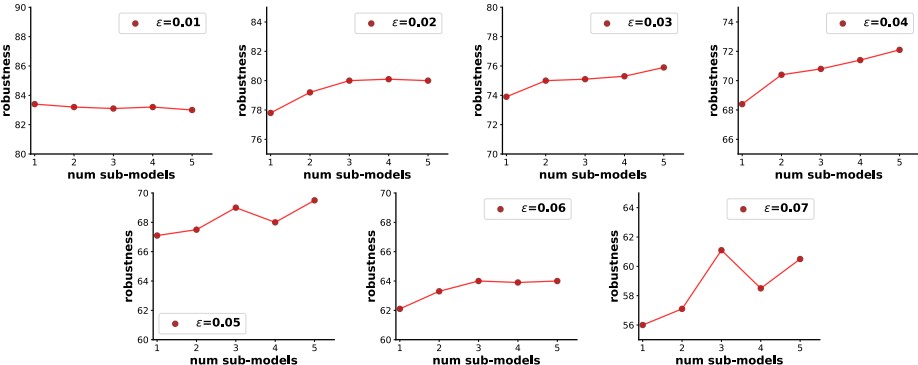

Figure 4: Robustness (%) using different number of sub-models under transfer attacks with 30 adversarial variants

necessary. When $0.02 \leq \epsilon \leq 0.03$, SoE achieves a better performance as the number of sub-models increases. If $\epsilon > 0.04$, SoE with multiple sub-models still outperforms a single model. However, SoE having $N = 3$ sub-models achieves a comparable robustness compared with SoE having $N = 5$. The volume of the sample surrounding is exponentially ($(1 + \epsilon)^{|\mathcal{X}|}$) large with the input $\mathcal{X}$ [2]. With the increase of $\epsilon$, the required model capacity has a sharp rise. We guess that given a large $\epsilon$ (e.g., $\epsilon = 0.07$), more sub-models (e.g., $N = 5$) may be insufficient to improve the performance further compared with learning SoE with $N = 3$. Similar phenomenon could also be found in Figure 4 when there are 30 generated adversarial variants.

## 4.4   Analysis about the Model Capacity of SoE

A model with sufficient capacity to cover all cases does not need to collaborate with others. To verify this claim, we conduct experiments using the ResNet model with different depths and show the clean accuracy (%) of single/multiple models in the following table.

Table 4: the accuracy using different model structures

| depth | 2 | 8 | 14 | 20 |
|---|---|---|---|---|
| single model | 65.0 | 88.3 | 90.5 | 91.9 |
| SoE | 67.0 | 89.5 | 91.6 | 92.5 |
| gain | 2.0 | 1.2 | 0.9 | 0.6 |

From the Table 4, with depth $= 2$, the model has the insufficient model capacity to learn the feature extractor, SoE can have a relatively large improvement (2.0). As the depth of the model is 20, the model has sufficient model capacity to fit all data samples. SoEachieves a slight improvement compared with a single model (0.6).

Compared with standard training, adversarial data are adaptively changed based on the current model to smooth the natural data's local neighborhoods. The volume of these surroundings is exponentially large. The model often encounters insufficient model capacity especially when there is a relatively large $\epsilon$ ball. Therefore, it is urgent to improve the utilization of the capacity for adversarial training.

## 5   Implementation Details

**the design of the E head**   In our experiments, we use ResNet20 as our backbone model. For the additional E head, we propose to use a one-layer MLP to model the relationship between the prediction and its confidence. Our design of the E head is based on the following aspects: 1). convenient optimization; a simple model structure can be optimized easily and will not bring a significant computation cost; 2). robustness; a complex E head may be vulnerable to the adversarial attack. Therefore, in our implementation, we choose to learn a simple E head for each sub-model to improve its robustness.

**Training devices**    We run our experiments on a local Linux server. We use Pytorch framework to implement our model and train all models on GeForce RTX 2080 Ti GPUs.