# OpenReview forum: "Synergy-of-Experts: Collaborate to Improve Adversarial Robustness"
_NeurIPS.cc/2022/Conference — NeurIPS 2022 Accept_

### Official Review · Reviewer_t6Y8 · 2022-07-10

**Rating:** 7
**Confidence:** 4
**Soundness:** 1 poor
**Presentation:** 3 good
**Contribution:** 2 fair

**Summary:**

This paper proposes an adversarial defense method that can better utilize the ensemble of multiple experts. In contrast to previous ensembling methods that require more than half methods give the correct prediction, for each sample, the proposed method employs a router module that routes the sample to the best model for prediction. In this way, the proposed method can yield the correct result if the selected model can predict right. The proposed method demonstrates better performance compared to previous ensemble methods.

**Questions:**

A more careful and convincing white attack method should be employed to evaluate the proposed method.

**Limitations:**

No social negative impact is found.

**Strengths And Weaknesses:**

Strengths:
1. The writing is good and easy to follow
2. There are some theoretical proofs

Weaknesses:
My main concern lies in the fairness of evaluation. For white-box attacks, I don't think the non-adaptive SoE can be a fair comparison: all adversarial methods assume the visibility of the whole model, which should include the E head. However, in non-adaptive SoE, E head is not visible.
The author also gives a setting with an adaptive adversarial attack that can attack the E head, which is fairer than the previous setting. However, I don't think the adaptive setting is fair enough. For example, besides attacking the best model to be less confident, all worse models should also be encouraged to predict higher confidence. I think this adversarial setting can be more challenging given the E head of worse models can be easier to be fooled for this image (since its backbone is easier to be fooled).
This example is one evaluation method. If the author can give a more systematic evaluation setting besides this, I would be more convincing.

Also, what's the adversarial sample employed in transfer adversarial attack? Is it adaptive or non-adaptive?

Some minor typos:
1. Algorithm 2, "duel" heads should be "dual" heads

---

> ### Author Response · Authors · 2022-08-02
> **Response to Reviewer t6Y8**
>
> We would like to thank the reviewer for the very insightful and valuable comments. Below are our responses to the comments.
> To the comments in ****Weaknesses:****
> * ****Questions: If the author can give a more systematic evaluation setting besides this, I would be more convincing.****
>
> ****Answer:**** we acknowledge that a fair comparison under white-box attack is crucial. In our experiments, we found that the attacks on a worse sub-model could have a lower success rate. More analysis and discussions are as follows.
>
> Firstly, we evaluate the robustness of the two heads of the best-performing sub-models.
>
> **1. only attacking the predictor of the best-performing sub-models;**
>
>  | $\epsilon$ | 0.01| 0.02 | 0.03 | 0.04 | 0.05 | 0.06 | 0.07|
>  | ----  | ---- | ---- | ---- | ---- | ---- | ---- | ---- |
>  | robustness | 72.0 | 57.5 | 47.8 | 38.7 | 30.4 | 24.3 | 24.0 |
>
> **2. only attacking the evaluator of the best-performing sub-models;**
>
> | $\epsilon$ | 0.01| 0.02 | 0.03 | 0.04 | 0.05 | 0.06 | 0.07|
>  | ----  | ---- | ---- | ---- | ---- | ---- | ---- | ---- |
>  | robustness | 71.2 | 63.2 | 51.2 | 47.6 | 40.4 | 36.1 | 35.3 |
>
> From the two tables above, attacking the predictor could achieve lower robustness. This could also be validated by the results shown in Figure 4 in the main text. From Figure 4, with  the  increase of $\lambda$, the attacker focuses  more on attacking the evaluator and our model has a higher robustness. Since we use a simple structure to implement the evaluator, attacking the evaluator could be harder than attacking the predictor to fool the collaboration.
>
> To verify the robustness of the evaluator of the worse sub-models, we maximize the loss $\ell = \mathrm{BCE}(g_{\phi}(x), 0)$ to increase $g_{\phi}(x)$.
>
> **3. only attacking the evaluator of the worse sub-models to increase the confidence;**
>
> | $\epsilon$ | 0.01| 0.02 | 0.03 | 0.04 | 0.05 | 0.06 | 0.07|
>  | ----  | ---- | ---- | ---- | ---- | ---- | ---- | ---- |
>  | robustness | 81.9 | 76.4 | 71.5 | 68.3 | 65.9 | 60.0 | 54.6 |
>
> From the above table, the collaboration has higher robustness when attacking the evaluator of the worse sub-models. For a best-performing sub-model, it may be susceptible because of overfitting. However, for a sub-model with poor performance, it may have a smoother decision boundary and maintains  a  more robust performance.
>
> We also conduct other adaptive methods which attack the predictor and the evaluator simultaneously using a weight loss with a proper $\lambda$.
>
> **4. attacking both the evaluator and the predictor of the best-performing sub-model;**
>
> | $\epsilon$ | 0.01| 0.02 | 0.03 | 0.04 | 0.05 | 0.06 | 0.07|
>  | ----  | ---- | ---- | ---- | ---- | ---- | ---- | ---- |
>  | robustness | 70.9 | 56.2 | 45.6 | 38.7 | 28.9 | 22.1 | 21.8 |
>
> **5. attacking both the evaluator and the predictor of the worse sub-models;**
>
> | $\epsilon$ | 0.01| 0.02 | 0.03 | 0.04 | 0.05 | 0.06 | 0.07|
>  | ----  | ---- | ---- | ---- | ---- | ---- | ---- | ---- |
>  | robustness | 74.3 | 62.2 | 55.0 | 46.2 | 38.6 | 29.8 | 27.1 |
>
> **6. attacking both the evaluator of the worse sub-model and the predictor of the best-performing sub-model;**
>
> | $\epsilon$ | 0.01| 0.02 | 0.03 | 0.04 | 0.05 | 0.06 | 0.07|
>  | ----  | ---- | ---- | ---- | ---- | ---- | ---- | ---- |
>  | robustness | 71.4 | 56.8 | 50.8 | 38.9 | 28.5 | 25.4 | 23.2 |
>
> From the results above, our method still performs better than baselines under various adaptive attacks.

---

> > ### Comment · Reviewer_t6Y8 · 2022-08-05
> > **Response to the authors**
> >
> > Thanks for the author's reply.
> >
> > I am also curious about attacking the predictor and evaluator of both worse sub-models and the best-performing sub-model.
> >
> > Also, one of my questions is not answered: "what's the adversarial sample employed in transfer adversarial attack? Is it adaptive or non-adaptive?"

---

> > > ### Author Response · Authors · 2022-08-06
> > > **Reply to the Reviewer t6Y8**
> > >
> > > Thanks for the reviewer's reply!
> > >
> > > * ****Question 1: I am also curious about attacking the predictor and evaluator of both worse sub-models and the best-performing sub-model.****
> > >
> > >  **Answer:** Thanks for your question. We would like to explain it as follows.
> > >
> > >  **7. attacking the predictor of both the best-performing sub-model and the worse sub-model.**
> > >
> > >  | $\epsilon$ | 0.01| 0.02 | 0.03 | 0.04 | 0.05 | 0.06 | 0.07|
> > >  | ---- | ---- | ---- | ---- | ---- | ---- | ---- | ---- |
> > >  | robustness | 73.2 | 61.4 | 52.3 | 43.5 | 34.0 | 27.6 | 25.5 |
> > >
> > >  **8. attacking the evaluator of both the best-performing sub-model and the worse sub-model.**
> > >
> > >  | $\epsilon$ | 0.01| 0.02 | 0.03 | 0.04 | 0.05 | 0.06 | 0.07|
> > >  | ---- | ---- | ---- | ---- | ---- | ---- | ---- | ---- |
> > >  | robustness | 83.3 | 63.7 | 63.0 | 54.3 | 51.1 | 46.9 | 45.6 |
> > >
> > >  From the above tables, the robustness of our method under the attack of the predictor (or the evaluator) of both the best-performing sub-model and the worse sub-model is higher than the robustness when only attacking the best-performing sub-model, and is lower when only attacking the worse sub-model.
> > >
> > >
> > >  **9. attacking the predictor and the evaluator of both the best-performing sub-model and the worse sub-model.** We also conduct experiments on attacking the dual heads of both the best-performing sub-model and the worse sub-model with an optimal $\lambda$. The results are shown in the following table. From the table, the robustness is slightly lower compared with the results in point **7**
> > >
> > >  | $\epsilon$ | 0.01| 0.02 | 0.03 | 0.04 | 0.05 | 0.06 | 0.07|
> > >  | ---- | ---- | ---- | ---- | ---- | ---- | ---- | ---- |
> > >  | robustness | 73.2 | 59.6 | 47.3 | 38.5 | 32.1 | 27.4 | 24.9 |
> > >
> > > From the tables above, our method still outperforms baselines under these adaptive attacks.
> > >
> > > * ****Question 2: Also, one of my questions is not answered: "what's the adversarial sample employed in transfer adversarial attack? Is it adaptive or non-adaptive?"****
> > >
> > >  **Answer:** We are sorry that we missed this problem when we went all out to answer the question about the adaptive attacks.
> > >
> > >  For the transfer adversarial attacks, we follow the setting in [1] as we stated in Lines 260-261 in our original submission. In particular, we use surrogate models to generate adversarial samples for all baselines. Since the surrogate model has only one predictor head, we can only attack the predictor head to generate transfer adversarial samples as [1] did. For each sample, we use various attack methods to generate adversarial variants. Only when the model can classify all kinds of adversarial variants can the model successfully defend against adversarial attacks. More information could be found in Sec 4.3 in the main text of our original submission.
> > >
> > > * ****Question 3: Some minor typos:..."****
> > >
> > >  **Answer:** We would like to thank the reviewer for pointing out the typos in our paper. We have corrected these typos and checked our paper carefully. The revised version of our submission has been uploaded.
> > >
> > > [1]. Huanrui Yang, Jingyang Zhang, Hongliang Dong, Nathan Inkawhich, Andrew Gardner, Andrew Touchet, Wesley Wilkes, Heath Berry, and Hai Li. Dverge: Diversifying vulnerabilities for enhanced robust generation of ensembles. In NeurIPS, 2020.

---

> > > > ### Comment · Reviewer_t6Y8 · 2022-08-06
> > > > **Response to authors**
> > > >
> > > > Thanks for your detailed reply. My concerns have been addressed and I would raise the score.

---

### Official Review · Reviewer_UDEK · 2022-07-11

**Rating:** 5
**Confidence:** 5
**Soundness:** 2 fair
**Presentation:** 3 good
**Contribution:** 2 fair

**Summary:**

This paper proposes a method to improve the adversarial robustness of an ensemble. Instead of using the voting strategy, the proposed method aims to identify the model with the highest confidence on the ground-truth label. In this way, it can improve the model performance when the correct prediction remains with the minority. The authors conduct experiments on both wight-box and black-box attacks to demonstrate the effectiveness of their method.

**Questions:**

- Given that $g_{\phi}$ and $f_{\theta}$ are learned alternatively, how stable is the learning?
- Which loss function is used in Table 1 and Figure 4? Why does the robustness increase when the weight $\lambda$ increases? Why is the robustness higher when $\epsilon=0.07$ than $\epsilon=0.06$?
- Have you measured the robustness when the ensemble contained different numbers of sub-models? When using SoE, can you use fewer sub-models to achieve higher robustness than previous voting-based methods?


**Limitations:**

The authors do not address the limitations of their work.

**Strengths And Weaknesses:**

[Strengths]
+ The authors rethink the classic voting-based strategy and note the problem with the case when correct predictions remain with the minority. Then, they propose a new “collaboration” strategy, which is very novel and interesting.
+ This paper is well-structured and easy to follow.

[Weaknesses]
- The formulation of the evaluator head in the SoE is questionable. In SoE, the evaluator head is designed to estimate the predicted probability $\hat p_y(x)$ through a BCE loss. However, if $g_{\phi}$ has successfully fitted the probability $\hat p_y(x)$, then we can directly identify which dimension in $\hat p(x)$ is the same as $g_{\phi}$, thereby obtaining the ground-truth label. Therefore, the ensemble seems useless.
- To learn the collaboration between sub-models, why not directly construct a one-hot label for the best-performing sub-model on each input, and use the cross-entropy to learn the probability of each sub-model being the best-performing model?
- The utility of the proposed method in minimizing the vulnerability overlap of all sub-models is not directly proven or verified. The main purpose of the proposed SoE is to minimize the vulnerability overlap of all sub-models. However, its effectiveness is not directly demonstrated. I suggest the authors provide some visual demonstrations or quantitative results to compare the vulnerability lap between SoE and other methods.
- In Algorithm 1, a surrogate loss is used to approximate the loss on the best-performing sub-model. Why not directly use the loss of the sub-model with the largest confidence $g_{\phi}(x)$?
- Algorithm 2 is somewhat confusing. The adversarial example is generated to worsen $\hat{p}(x)$, which is the output of the sub-model with the highest confidence. Therefore, this adversarial example is generated only based on the best-performing sub-model. Actually, such adversarial examples have been used in Algorithm 1, so it is confusing why we need Algorithm 2.
- The attacking methods used in experiments are not sufficient. First, the proposed method should be at least compared with the PGD/C&W attack on the best-performing sub-model. Second, the attack that simultaneously destroys the estimator output and the predictor head should be considered. Third, the adaptive attack used in the paper is confusing. There should be more discussions about $l_2$ and the selection of $j$. Besides, in $l_2$, $y$ is usually not known by the attacker, so it is unclear how to compute $l_2$.

---

> ### Author Response · Authors · 2022-08-02
> **Part 1**
>
>
> We would like to thank the reviewer for appreciating our novelty and the very valuable comments. Below are our response to the comments.
>
> To the comments in ****Weaknesses:****
> * ****Question 1: The formulation of the evaluator head in the SoE is questionable. In SoE, the evaluator head is designed to estimate the predicted probability $\hat{p}\_{y}(x)$ through a BCE loss. However, if $g\_{\phi}$ has successfully fitted the probability $\hat{p}\_{y}(x)$ then we can directly identify which dimension in $\hat{p}\_{y}(x)$ is the same as $g\_{\phi}$ thereby obtaining the ground-truth label. Therefore, the ensemble seems useless.****
>
>  ****Answer:**** we would like to clarify it and the reasons are as follows:
>
>  (1). ****it is almost impossible to learn such a perfect evaluator;**** the performance of the evaluator is largely affected by the feature extractor. However, in adversarial training, the feature extractor is hard to fit all adversarial samples when there is a relatively large $\epsilon$. Therefore, the evaluator can be hardly perfect;
>
>  (2). ****given an effective evaluator, we still may not directly identify the ground-truth label.**** Since a deep model usually confidently predicts the label is a specific class, for example, $\boldsymbol{\hat{p}} = [0.91, 0.008,0.01,0.009,0.011,0.007,0.01,0.011,0.013,0.011]$. Suppose the predicted confidence is 0.01 ($g_{\phi} = 0.01$), it is hard to identify the ground-truth label by identifying which dimension in $\boldsymbol{\hat{p}}$ is the same as $g_{\phi}$.
>
>  (3). ****the collaboration mechanism aims to minimize the vulnerability overlap during training, and the evaluator focuses on identifying the prediction with the highest confidence during inference.**** In our proposed framework, the collaboration mechanism means we train sub-models collaboratively so that it could achieve a smaller vulnerability area. During inference, our framework outputs the predictions with the highest confidence. Therefore, there is no collaboration between sub-models during inference as the reviewer mentioned.
>
> * ****Question 2: To learn the collaboration between sub-models, why not directly construct a one-hot label for the best-performing sub-model on each input, and use the cross-entropy to learn the probability of each sub-model being the best-performing model?****
>
> ****Answer:**** Thanks for your question. We would like to explain it from the following aspects:
>
> (1). ****the confidence is more informative than one-hot label.**** the confidence which refers to the predicted label probability is more informative than one-hot label, when evaluating the quality of the predictions. For example, suppose there are three sub-models, when the predicted label probability of the three sub-models are $\hat{p}\_{y}^{1} = 0.9$, $\hat{p}\_{y}^{2} = 0.1$, $\hat{p}\_{y}^{3} = 0.1$, the one-hot label should be [1, 0, 0]. However, if the predicted label probability of the three sub-models are $\hat{p}\_{y}^{1} = 0.9$, $\hat{p}\_{y}^{2} = 0.89$, $\hat{p}\_{y}^{3} = 0.1$, the one-hot labels will still be [1, 0, 0]. In this case, if we use one-hot label to learn models, it will be harder to find the suboptimal predictions ($\hat{p}\_{y}^{2} = 0.89$) once the best-performing sub-model is attacked.
>
> (2). ****learning one-hot label may be more difficult to optimize**** In fact, we have tried one-hot learning to identify the best-performing sub-model, but the experiments show that it has lower robustness. Suppose there are three sub-models, the one-hot label of the sub-model $f^{1}$ is determined by the predictions of other sub-models ($f^{2}$ and $f^{3}$). If other sub-models perform worse, the one-hot label of $f^{1}$ is 1, otherwise it is 0. In this case, the one-hot label predictor uses all predictions as input, e.g., $g_{\phi}(f^{1}(x), f^{2}(x), f^{3}(x))$. We guess the reason behind the poor performance is that the evaluator may be too complex to optimize.

---

> ### Author Response · Authors · 2022-08-02
> **Part 2**
>
> * ****Question 3: The utility of the proposed method in minimizing the vulnerability overlap of all sub-models is not directly proven or verified. The main purpose of the proposed SoE is to minimize the vulnerability overlap of all sub-models. However, its effectiveness is not directly demonstrated. I suggest the authors provide some visual demonstrations or quantitative results to compare the vulnerability lap between SoE and other methods.****
>
> ****Answer:**** We thank the reviewer for the advice on directly verifying our proposed collaboration scheme. We would like to explain it from the following aspects:
>
> 1. ****visual demonstration:**** following the suggestion, we provide a visualization of the vulnerability area of the collaboration and the ensemble. We add it in Sec. C.2 in Appendix and highlight it in blue. Compared with the ensemble which requires the sub-models to fit the same adversarial samples to defend against adversarial attacks, the collaboration encourages more diverse vulnerability areas of sub-models to minimize the vulnerability overlap. From Figure 7 in Appendix, our collaboration fixes a  broader vulnerability area than the ensemble. More details could be found in Sec. C.2 in our rebuttal submission;
>
> 2. ****quantitative results:**** The black-box attack is a possible method to quantify the vulnerability overlap. In our original submission, we did black-box attack experiments to validate the effectiveness of the collaboration scheme. The results in the following table are copied from the original submission.
>
> black-box attack experiments
>
>  | $\epsilon$ | 0.01| 0.02 | 0.03 | 0.04 | 0.05 | 0.06 | 0.07|
>  | ----  | ---- | ---- | ---- | ---- | ---- | ---- | ---- |
>  | GAL | 47.0$_{\pm 2.3}$ | 43.8$_{\pm1.8}$ | 26.4$_{\pm 3.2}$ | 27.7$_{\pm 2.1}$ | 18.5$_{\pm 1.5}$  | 13.1$_{\pm 1.2}$ | 8.40$_{\pm 2.4}$|
>  | DVERGE |  72.0$_{\pm 1.2}$ | 58.8$_{\pm 1.1}$ | 47.8$_{\pm 1.0}$ | 37.9$_{\pm 1.1}$ | 28.4$_{\pm 1.0}$ | 21.0$_{\pm 1.2}$ | 15.0$_{\pm 1.1}$|
>  | ADP  |  72.5$_{\pm 1.0}$  |60.3$_{\pm 1.1}$ | 47.2$_{\pm 1.3}$ | 37.9$_{\pm 1.4}$ | 28.0$_{\pm 1.3}$ | 25.5$_{\pm 1.0}$ | ****21.3****$_{\pm 1.2}$ |
>  | MoRE  |  72.8$_{\pm .8}$ | 59.6$_{\pm .8}$ | 46.4$_{\pm 1.1}$ | 37.8$_{\pm 1.2}$ | 30.1$_{\pm 1.3}$ | 22.0$_{\pm 1.7}$ | 16.6$_{\pm 1.9}$ |
>  | SoE | ****77.7****$_{\pm .5}$ | ****65.1****$_{\pm 1.0}$ | ****54.7****$_{\pm 1.0}$ | ****45.3****$_{\pm 1.0}$ | ****36.3****$_{\pm 1.3}$ | ****29.2****$_{\pm 1.3}$ | 18.0$_{\pm 1.5}$|
>
> From the table, our method achieves better robustness compared with the ensemble methods, and this verifies that our collaboration scheme could achieve a smaller vulnerability overlap.
>
> * ****Question 4: In Algorithm 1, a surrogate loss is used to approximate the loss on the best-performing sub-model. Why not directly use the loss of the sub-model with the largest confidence $g\_{\phi}(x)$?****
>
> ****Answer:**** Thanks for the question. We would like to explain it from the following aspects:
>
> (1). our collaboration aims to fit the adversarial samples using the true best-performing sub-models. However, at the beginning of the model training, $g\_{\phi}(x)$ may not effectively identify the best-performing sub-model. Therefore, we do not directly use the loss of the sub-model with the largest confidence $g\_{\phi}(x)$;
>
> (2). directly optimizing the objective of the sub-model with the highest confidence may cause a potential trivial case. For example, if we only optimize the sub-model with the highest confidence, there may be only one sub-model that is properly trained. Therefore, we optimize a surrogate loss to minimize the objectives of all sub-models.

---

> ### Author Response · Authors · 2022-08-02
> **Part 3**
>
> * ****Question 5: Algorithm 2 is somewhat confusing....****
>
> ****Answer:**** we use Alg. 1 and Alg. 2 in sequence to train our collaboration. And Alg. 2 provides unseen adversarial samples. The reasons are as follows.
>
> In Alg 1, the generated samples $\tilde{x}$ are the ****most**** adversarial samples of sub-models. The samples generated by one sub-models are fit by other sub-models with the best performance. Therefore, after the training converges, there are adversarial samples that could be ****not the most**** adversarial samples of any sub-model, but could fool all sub-models.
>
> Alg 2 is proposed to obtain such adversarial samples. In Alg 2, $\tilde{x}''$ are generated to worsen $\hat{p}$, which is the output of the sub-model with the highest confidence. However, during the generalization of $\tilde{x}''$, given the input $x$, we could firstly obtain an adversarial sample $\tilde{x}'$ which could fool the best-performing model. Then we continue to generate $\tilde{x}''$ to fool another sub-model with the best performance on $\tilde{x}'$.
>
> We did ablation studies to validate the effectiveness of Alg. 2 in our original submission. We copy the results for your easy reference.
>
>  | $\epsilon$ | 0.01| 0.02 | 0.03 | 0.04 | 0.05 | 0.06 | 0.07|
>  | ----  | ---- | ---- | ---- | ---- | ---- | ---- | ---- |
>  | SoE (without Alg. 2) | 85.0 | 82.3 | 78.5 | 75.0 | 65.0  | 50.0 | 30.0 |
>  | SoE |  85.2 | 83.4 | 78.8 | 76.6 | 74.6 | 72.3 | 70.2 |
>
> From the above table, without Alg. 2, the robustness of SoE decreases dramatically when $\epsilon > 0.04$. More discussion could be found in Sec. 4.4 in our original submission.
>
> * ****Question 6: The attacking methods used in experiments are not sufficient. First, the proposed method should be at least compared with the PGD/C&W attack on the best-performing sub-model. Second, the attack that simultaneously destroys the estimator output and the predictor head should be considered. Third, the adaptive attack used in the paper is confusing. There should be more discussions about $l_{2}$ and the selection of $j$. Besides, in $l_{2}$, $y$ is usually not known by the attacker, so it is unclear how to compute $l_{2}$.****
>
>  ****Answer:**** We would like to explain the three points as follows.
>
>  ****(1).**** in our original submission, we did C&W attack in transfer attack experiments and black-box attacks in Sec 4.1 in Appendix. In our white-box attack experiments, we only used PGD attacks. Following the advice, we conduct experiments under white-box attacks using C&W attack and the results are shown in the following table.
>
>  | $\epsilon$ (robust/clean) | 0.01| 0.02 | 0.03 | 0.04 | 0.05 | 0.06 | 0.07|
>  | ----  | ---- | ---- | ---- | ---- | ---- | ---- | ---- |
>  | GAL | 41.5/87.8 | 36.4/85.4 | 21.381.2 | 22.9/78.7 | 16.0/77.3 | 9.9/76.2 | 6.4/76.0 |
>  | DVERGE | 67.1/85.4 | 51.2/83.0 | 39.3/79.7 | 29.5/77.6 | 21.6/76.7 | 14.9/75.8 | 9.3/75.3 |
>  | ADP  | 66.9/89.0 | 51.5/86.8 | 38.9/85.4 | 28.9/83.3 | 21.9/76.0 | 20.9/66.4 | 15.2/63.0 |
>  | MoRE  | 65.3/88.0 | 50.7/85.3 | 37.4/82.0 | 30.8/79.5 | 22.9/78.2 | 15.6/77.1 | 11.2/77.8 |
>  | SoE | **70.4**/88.8 | **55.4**/85.6 | **43.4**/80.2 | **33.6**/80.0 | **25.2**/79.1 | **19.6**/76.7 | **15.6**/74.1 |
>
> Moreover, we conduct auto-attack and the results are shown in the following table.
>
>  | $\epsilon$ (robust/clean) | 0.01| 0.02 | 0.03 | 0.04 | 0.05 | 0.06 | 0.07|
>  | ----  | ---- | ---- | ---- | ---- | ---- | ---- | ---- |
>  | GAL | 39.0/87.8 | 34.1/85.4 | 18.5/81.2 | 20.2/78.7 | 12.2/77.3 | 7.2/76.2 | 4.7/76.0 |
>  | DVERGE | ****69.0****/85.4 | 52.3/83.0 | 36.9/79.7 | 28.7/77.6 | 19.7/76.7 | 10.8/75.8 | 7.2/75.3 |
>  | ADP  | 66.6/89.0 | 51.6/86.8 | 39.6/85.4 | 27.7/83.3 | 17.6/76.0 | 11.5/66.4 | 7.3/63.0 |
>  | MoRE  | 64.1/88.0 | 49.9/85.3 | 35.9/82.0 | 28.0/79.5 | 19.3/78.2 | 11.9/77.1 | 7.0/77.8 |
>  | SoE | 68.4/88.8 | ****53.4****/85.6 | ****42.3****/80.2 | ****32.5****/80.0 | ****22.9****/79.1 | ****13.0****/76.7 | ****7.5****/74.1|
>
> From the above two tables, our method still outperforms baselines under C&W attack and auto-attack.
>
> ****(2).**** In our original submission, we did two different adaptive attacks to attack the predictor and the estimator by maximizing a weighted loss. We show the results under the strongest adaptive attacks in Table 1. The experiments show that adaptive attacks slightly degrade the robustness of our method but our method still outperforms baselines.
>
> ****(3). adaptive attack is white-box attack. The attacker should know the label $y$ under white-box attack.**** From the formulation of $\ell\_{2}$, the gradient of $f_{\theta}(x)\_{y}$ is always negative, while the gradient of $g\_{\phi}(x)$ is always positive.  Maximizing $\ell\_{2}$ could decrease the predicted label probability ($f_{\theta}(x)\_{y}$) and increase the confidence ($g\_{\phi}(x)$). Therefore, maximizing $\ell\_{2}$ encourages a mismatch between the predicted probability $f_{\theta}(x)\_{y}$ and the confidence $g\_{\phi}(x)$.

---

> ### Author Response · Authors · 2022-08-02
> **Part 4**
>
> To the comments in ****Questions:****
>
> * ****Question 7: Given that $g\_{\phi}$ and $f\_{\theta}$ are learned alternatively, how stable is the learning?.****
>
> ****Answer:**** Instead of learning $g\_{\phi}$ and $f\_{\theta}$ alternatively, we learn $f\_{\theta}$ and $g_{\phi}$ simultaneously. From Figure 2(b) in the main text, $f\_{\theta}$ contains the parameters of the feature extractor and the predictor. $g\_{\phi}$ only contains the parameters of the evaluator. Optimizing $g\_{\phi}$ and $f\_{\theta}$ could update different parameters. Therefore, our method is stable as baselines.
>
> * ****Question 8: Which loss function is used in Table 1 and Figure 4? Why does the robustness increase when the weight $\lambda$ increases? Why is the robustness higher when $\epsilon = 0.07$ than $\epsilon = 0.06$ ?****
>
> ****Answer:**** (1). we use two adaptive attacks with two different loss functions ($\ell^{adp}\_{1}$ and $\ell^{adp}\_{2}$). In table 1, we show the robustness under the strongest loss functions with an optimal $\lambda$. In Figure 4, we also show the results of the loss function which achieves a stronger attack under different $\lambda$.
>
> (2). In our experiments, we found that the evaluator is more difficult to be successfully attacked compared with the predictor due to its simple structure. With the increase of $\lambda$, the attacker focuses more on fooling the evaluator rather than the predictor and this leads to a lower attack success rate.
>
> (3). the robustness is indeed lower when $\epsilon = 0.07$ than $\epsilon = 0.06$. All methods achieve a similar robustness when $\epsilon = 0.06$ and $\epsilon = 0.07$. From Figure (4), the robustness is lower when $\epsilon = 0.07$ (24.0%) than $\epsilon = 0.06$ (24.3%) if we only attack the predictor ($\lambda = 0.0$). Besides, if we select an optimal $\lambda$, the robustness is still slightly lower when $\epsilon = 0.07$ (21.8%) than $\epsilon = 0.06$ (22.1%).
>
> * ****Question 9: Have you measured the robustness when the ensemble contained different numbers of sub-models? When using SoE, can you use fewer sub-models to achieve higher robustness than previous voting-based methods?****
>
> ****Answer:**** we did analyze the affect of the number of sub-models on the robustness in Sec 4.2 in Appendix in our original submission. In summary, we have the following findings:
>
> (1). ****different $\epsilon$:**** multiple sub-models could achieve a higher robustness improvement given a relatively  large $\epsilon$.
>
> (2). ****different number of sub-models:**** more sub-models are more likely to achieve higher robustness, but the margin gain decreases with more sub-models.
>
> For the comparison with voting-based methods when using fewer sub-models, we copy the results under transfer attacks in which we use 2 sub-models of SoE.
>
>  | $\epsilon$ | 0.01| 0.02 | 0.03 | 0.04 | 0.05 | 0.06 | 0.07|
>  | ----  | ---- | ---- | ---- | ---- | ---- | ---- | ---- |
>  | GAL (3 sub-models) | 64.2$_{\pm 4.2}$ | 48.7$_{\pm 2.7}$ | 50.2$_{\pm 3.5}$ | 49.9$_{\pm 3.2}$ | 52.3$_{\pm 4.5}$ | 48.7$_{\pm 3.2}$ | 42.2$_{\pm 4.1}$ |
>  | ADP (3 sub-models)  | 85.6$_{\pm.2}$  | 82.9$_{\pm .2}$ | 78.3$_{\pm .3}$ | 73.2$_{\pm .1}$ | 69.6$_{\pm .2}$ | 60.4$_{\pm .2}$ | 57.4$_{\pm .1}$ |
>  | DVERGE (3 sub-models)  | 83.4$_{\pm .3}$ | 80.1$_{\pm .2}$ | 77.3$_{\pm .1}$ | 72.4$_{\pm .1}$ | 71.9$_{\pm .2}$ | 68.8$_{\pm .3}$ | 66.2$_{\pm .2}$ |
>  | MoRE (3 sub-models) |   84.8$_{\pm .3}$ | 82.1$_{\pm .1}$ | 78.4$_{\pm .2}$ | 74.3$_{\pm .1}$ | 73.2$_{\pm .1}$ | 70.3$_{\pm .2}$ | 69.1$_{\pm .3}$ |
>  | SoE (2 sub-models) |  85.0$_{\pm .1}$  | 82.5$_{\pm .2}$ | 78.0$_{\pm .1}$ | 74.2$_{\pm .1}$ | 73.0$_{\pm .1}$ |70.5$_{\pm .2}$ |64.2$_{\pm .2}$ |
>
> From the above table, SoE with two sub-models achieves similar robustness compared with voting-based methods with three sub-models, which means a more efficient utilization of the sub-models of our method. More discussion could be found in Sec 4.2 in Appendix in our original submission.

---

> ### Author Response · Authors · 2022-08-06
> **Need Further Clarification?**
>
> Dear reviewer UDEK:
>
> We appreciate your efforts in reviewing our paper. Would you mind checking our response, and is there any unclear point so that we could further clarify?
>
> Best regards,
>
> Authors

---

> > ### Comment · Reviewer_UDEK · 2022-08-08
> > **Response to the authors**
> >
> > Thank you for your great efforts, and most of my concerns have been addressed. Some details are still not clear.
> >
> > First, the generation of adversarial examples $\tilde{x}''$ in Part 3 of your response is not discussed in the paper. I'm confused that whether this is what Algorithm 2 exactly describes, or just an optional setting in the implementation of Algorithm 2. I prefer the authors to clarify this setting.
> >
> > Second, it seems that Table 1 was obtained by using the PGD attack, instead of $\ell_1^{adp}$ or $\ell_2^{adp}$ in your response. My question is, which submodel was used when computing the loss in the PGD attack? Did you use the loss function of the best-performing model or the sum of losses of all submodels? Similarly, in $\ell_1^{adp}$ and $\ell_2^{adp}$, it is unclear that the first term $\ell(f_{\theta}(x),y)$ was computed on which submodel.
> >
> > Besides, I suggest the authors put some interesting results to the main text, *e.g.,* the visualization in Appendix C.2 and the discussion about the number of submodels in Section 4.2 in Appendix. These results would better demonstrate the effectiveness of the proposed method.

---

> > > ### Author Response · Authors · 2022-08-08
> > > **Reply to Reviewer UDEK**
> > >
> > > Thanks for your reply! We would like to make every effort to clarify the unclear points.
> > >
> > > **Question 1:First, the generation of adversarial examples $\tilde{x}''$ in Part 3 of your response is not discussed in the paper. I'm confused that whether this is what Algorithm 2 exactly describes, or just an optional setting in the implementation of Algorithm 2. I prefer the authors to clarify this setting.**
> > >
> > > **Answer:** As the reviewer mentioned, the generalization process of $\tilde{x}''$ in Part 3 is what Algorithm 2 exactly describes. During adversarial training, we generate adversarial samples using the PGD attack iteratively. In Algorithm 2, we attack the collaboration iteratively to obtain the adversarial samples. However, given a benign sample $x$, when we update the adversarial sample in each iteration, the best-performing sub-model could be different. As we stated in Part 3, in the first iteration, given the input $x$, the best-performing sub-model is $f_{1}$. We obtain an adversarial sample $\tilde{x}'$ by attacking $f_{1}$. However, in the second iteration, given the input $\tilde{x}'$, the best-performing sub-model is another sub-model (e.g., $f_{2}$) rather than $f_{1}$. We attack the best-performing sub-model $f_{2}$ to obtain the adversarial sample $\tilde{x}''$. Therefore, by attacking the collaboration following Algorithm 2, we could obtain the adversarial sample $\tilde{x}''$ stated in Part 3.
> > >
> > > In our original submission, we only stated that Algorithm 2 could generate the unseen adversarial samples. Following the advice, we have added this detail to our revised submission and highlighted it in blue.
> > >
> > > **Question 2: Second, it seems that Table 1 was obtained by using the PGD attack, instead of $\ell_{1}^{adp}$ or $\ell_{2}^{adp}$ in your response.**
> > >
> > > **Answer:** We would like to explain that in Table 1, the second to the last line **SoE** is the robustness results under the PGD attack, and the last line **SoE (adaptive)** is the robustness results by using the loss function $\ell_{1}^{adp}$ or $\ell_{2}^{adp}$.
> > >
> > > **Question 3: My question is, which submodel was used when computing the loss in the PGD attack? Did you use the loss function of the best-performing model or the sum of losses of all submodels? Similarly, in $l_{1}^{adp}$ and $l_{2}^{adp}$, it is unclear that the first term $\ell(f_{\theta}(x), y)$ was computed on which submodel.**
> > >
> > > **Answer:** We would like to explain it as follows.
> > >
> > > (1). maximizing the loss $\ell(f_{\theta}(x), y)$ could worse the predictor head, and maximizing the loss $l_{1}^{adp}$ ($l_{2}^{adp}$) could worse the evaluator.
> > >
> > > (2). we use a weighted loss $\ell(f_{\theta}(x), y) + \lambda \cdot l_{1}^{adp}$. By maximizing this weighted loss by the PGD attack, we could attack the predictor and the evaluator simultaneously.
> > >
> > > (3). in our original experiments under adaptive attacks, we attack the best-performing sub-model, which means we use the loss functions of the best-performing model rather than the sum of losses of all sub-models. Therefore, when computing the first term $\ell(f_{\theta}(x), y)$, we also use the loss of the best-performing sub-model to attack the predictor of the best-performing sub-model.
> > >
> > > (4). following the advice from Reviewer t6Y8, we also conduct experiments by attacking the predictor (evaluator) of the worse sub-models. Experiments under 9 different adaptive attacks validate that our method could outperform baselines on white-box attacks. More details could be found in **Response to Reviewer t6Y8**.
> > >
> > > **Question 4: Besides, I suggest the authors put some interesting results to the main text, e.g., the visualization in Appendix C.2 and the discussion about the number of submodels in Section 4.2 in Appendix. These results would better demonstrate the effectiveness of the proposed method.**
> > >
> > > **Answer:** We would like to thank the reviewer for the kind advice. Due to the page limit, we put these results in Appendix in our last submission. Following the advice, we tried our best to put these contents in the main text and highlighted them in blue. Meanwhile, due to the page limit, some details were left in Appendix. The revised submission has been uploaded.

---

> > > > ### Comment · Reviewer_UDEK · 2022-08-09
> > > > **Response to authors**
> > > >
> > > > Thank you for your clarification. I do not have other questions, and I would raise the score.

---

### Official Review · Reviewer_1A7M · 2022-07-11

**Rating:** 7
**Confidence:** 3
**Soundness:** 3 good
**Presentation:** 3 good
**Contribution:** 3 good

**Summary:**

This work introduces a model collaboration scheme to tackle the insufficient model capacity against adversarial examples instead of model ensemble. Through replacing voting-based strategy with selecting the best-performing sub-model, each sub-model only fits its specific adversarial areas, which enables the models with limited capacity to achieve better adversarial robustness. An auxiliary head which outputs the confidence is introduced to identify the best-performing sub-models. Experiments on CIFAR-10 and ResNet-20 demonstrate the effectiveness of proposed algorithm.

**Questions:**

Overall, I think this paper is interesting. I have several suggestions for the authors:

1. Please provide some empirical evidence that sub-model handles its corresponding adversarial subspace via collaboration scheme.
2. Please include some stronger attacks for comparison, such as autoattack.


**Limitations:**

The authors mentioned that they have discussed the limitations in the Appendix.

**Strengths And Weaknesses:**

Pros:

* This work is well-organized with clear motivation and generally easy to read.
* The authors provide the source code with instructions, which makes the algorithm easy to follow.
* The designing of collaboration scheme seems natural and reasonable with its corresponding theoretical analysis.
* The comparison with other defense baselines on CIFAR-10 demonstrate the superiority of proposed algorithm.

Cons:

* The motivation of this paper is that each sub-model can handle its corresponding adversarial subspace via collaboration scheme so that the adversarial robustness with limited model capacity can be improved. However, it is difficult to the empirical evidence of it in experimental section. For example, some visualization of adversarial examples with decision boundaries of collaborated models are recommended.
* The performance under different attack settings includes PGD and adaptive attacks is reported. However, some popular stronger attacks are not included for comparison, such as autoattack.

---

> ### Author Response · Authors · 2022-08-02
> **Response to Reviewer 1A7M**
>
> We would like to thank the reviewer for the positive and very valuable comments. Below are our responses to the comments in ****Questions****.
>
> * ****Question 1: Please provide some empirical evidence that sub-model handles its corresponding adversarial subspace via collaboration scheme.****
>
>  ****Answer:**** we would like to explain it from the following two aspects:
> 1. ****visualization:**** following the suggestion, we provide a visualization of adversarial samples with the decision boundaries of collaborated models. We add it in Sec. C.2 in Appendix and highlight it in blue. We compare the vulnerability of the collaboration and the ensemble scheme. From Figure 7 in Appendix, our collaboration achieves a smaller vulnerability area than the ensemble. More details could be found in Sec. C.2 in our rebuttal submission;
> 2. ****quantitive validation**** The black-box attack is a possible method to quantify the vulnerability area. In our original submission, we did black-box attack experiments to validate the effectiveness of the collaboration scheme. The results are copied from the original submission.
>
>  | $\epsilon$ | 0.01| 0.02 | 0.03 | 0.04 | 0.05 | 0.06 | 0.07|
>  | ----  | ---- | ---- | ---- | ---- | ---- | ---- | ---- |
>  | GAL | 47.0$_{\pm 2.3}$ | 43.8$_{\pm1.8}$ | 26.4$_{\pm 3.2}$ | 27.7$_{\pm 2.1}$ | 18.5$_{\pm 1.5}$  | 13.1$_{\pm 1.2}$ | 8.40$_{\pm 2.4}$|
>  | DVERGE |  72.0$_{\pm 1.2}$ | 58.8$_{\pm 1.1}$ | 47.8$_{\pm 1.0}$ | 37.9$_{\pm 1.1}$ | 28.4$_{\pm 1.0}$ | 21.0$_{\pm 1.2}$ | 15.0$_{\pm 1.1}$|
>  | ADP  |  72.5$_{\pm 1.0}$  |60.3$_{\pm 1.1}$ | 47.2$_{\pm 1.3}$ | 37.9$_{\pm 1.4}$ | 28.0$_{\pm 1.3}$ | 25.5$_{\pm 1.0}$ | ****21.3****$_{\pm 1.2}$ |
>  | MoRE  |  72.8$_{\pm .8}$ | 59.6$_{\pm .8}$ | 46.4$_{\pm 1.1}$ | 37.8$_{\pm 1.2}$ | 30.1$_{\pm 1.3}$ | 22.0$_{\pm 1.7}$ | 16.6$_{\pm 1.9}$ |
>  | SoE | ****77.7****$_{\pm .5}$ | ****65.1****$_{\pm 1.0}$ | ****54.7****$_{\pm 1.0}$ | ****45.3****$_{\pm 1.0}$ | ****36.3****$_{\pm 1.3}$ | ****29.2****$_{\pm 1.3}$ | 18.0$_{\pm 1.5}$|
>
> From the table, our method achieves better robustness compared with the ensemble methods, and this validates that our collaboration could fix broader vulnerability areas.
>
> * ****Question 2: Please include some stronger attacks for comparison, such as auto attack.****
>
> ****Answer:**** following the suggestion, we compare our method with baselines under auto-attack. The results are shown in the following table.
>
>  | $\epsilon$ (robust/clean) | 0.01| 0.02 | 0.03 | 0.04 | 0.05 | 0.06 | 0.07|
>  | ----  | ---- | ---- | ---- | ---- | ---- | ---- | ---- |
>  | GAL | 39.0/87.8 | 34.1/85.4 | 18.5/81.2 | 20.2/78.7 | 12.2/77.3 | 7.2/76.2 | 4.7/76.0 |
>  | DVERGE | ****69.0****/85.4 | 52.3/83.0 | 36.9/79.7 | 28.7/77.6 | 19.7/76.7 | 10.8/75.8 | 7.2/75.3 |
>  | ADP  | 66.6/89.0 | 51.6/86.8 | 39.6/85.4 | 27.7/83.3 | 17.6/76.0 | 11.5/66.4 | 7.3/63.0 |
>  | MoRE  | 64.1/88.0 | 49.9/85.3 | 35.9/82.0 | 28.0/79.5 | 19.3/78.2 | 11.9/77.1 | 7.0/77.8 |
>  | SoE | 68.4/88.8 | ****53.4****/85.6 | ****42.3****/80.2 | ****32.5****/80.0 | ****22.9****/79.1 | ****13.0****/76.7 | ****7.5****/74.1|
>
> From the table above, auto-attack is stronger than PGD and it achieves a higher attack success rate. Our method still outperforms baselines when $\epsilon > 0.01$.

---

### Meta-Review · Area_Chair_cAN7 · 2022-08-25

**Recommendation:** Accept
**Confidence:** Certain

**Metareview:**

This paper proposes an ensemble-type solution for improving the adversarial robustness of a model. The proposed idea is simple, yet novel with theoretical supports. The authors did a good job clarifying reviewers' concerns and all reviewers finally recommend acceptance. AC also thinks that this is a good paper in various aspects (novel idea, good write-up, solid theoretical supports) and has a potential to be a generic ensemble-type solution even in non-adversarial/standard setups, e.g., see [1]. Furthermore, the confidence prediction can be used for other purposes, e.g., out-of-distribution detection [2] and active learning [3]. It is useful for readers to discuss about these extensions.

[1] Confident Multiple Choice Learning, Kimin Lee et al., ICML 2017.

[2] Out-of-Distribution Detection Using an Ensemble of Self Supervised Leave-out Classifiers, Apoorv Vyas et al., ECCV 2018.

[3] Learning Loss for Active Learning, Donggeun Yoo and In So Kweon, CVPR 2019.

**Award:**

No

---

### Decision · Program_Chairs · 2022-09-14

Accept